# A BIRD'S EYE VIEW ON INFORMED CLASSIFICATION

## ABSTRACT

Neurosymbolic AI is a growing field of research aiming to combine neural network learning capabilities with the reasoning abilities of symbolic systems. In this paper, we tackle informed classification tasks, *i.e.*, multi-label classification tasks informed by prior knowledge that specifies which combinations of labels are semantically valid. Several neurosymbolic formalisms and techniques have been introduced in the literature, each relying on a particular language to represent prior knowledge. We take a bird's eye view on informed classification and introduce a unified formalism that encapsulates all knowledge representation languages. Then, we build upon this formalism to identify several concepts in probabilistic reasoning that are at the core of many techniques across representation languages. We also define a new technique called semantic conditioning at inference, which only constrains the system during inference while leaving the training unaffected, an interesting property in the era of off-the-shelves and foundation models. We discuss its theoritical and practical advantages over two other probabilistic neurosymbolic techniques: semantic conditioning and semantic regularization. We then evaluate experimentally and compare the benefits of all three techniques on several large-scale datasets. Our results show that, despite only working at inference, our technique can efficiently leverage prior knowledge to build more accurate neural-based systems.

## 1 INTRODUCTION

Neurosymbolic AI is a growing field of research aiming to combine neural network learning capabilities with the reasoning abilities of symbolic systems. This hybridization can take many shapes depending on how the neural and symbolic components interact (Kautz, 2022; Wang et al., 2023). An important sub-field of neurosymbolic AI is Informed Machine Learning (von Rueden et al., 2023), which studies how to leverage prior knowledge to improve neural-based systems. There again, proposed techniques in the literature can be of very different nature depending on the type of task (*e.g.* regression, classification, detection, generation, etc.), the formalism used to represent the prior knowledge (*e.g.* mathematical equations, knowledge graphs, logical theories, etc.), the stage at which knowledge is embedded (*e.g.* data processing, neural architecture design, learning procedure, inference procedure, etc.) and benefits expected from the hybridization (*e.g.* explainability, performance, frugality, etc.).

In particular, informed classification studies multi-label classification tasks where prior knowledge specifies which combinations of labels are semantically valid. In our work, the architecture of the neural model (*e.g.* fully connected, convolutional, transformer-based, etc.) mainly depends on the modality of the input space (*e.g.* images, texts, etc.). Therefore, we consider model-agnostic neurosymbolic techniques that integrate prior knowledge during learning, inference or both, but leave the design of the architecture outside the reach of the technique. To lighten our formalism, we restrict ourselves to the supervised setting throughout the paper, even though some of the techniques we mention can be used in a semi-supervised setting. Several neurosymbolic formalisms and techniques have been introduced in the literature, each using a particular language to represent prior knowledge: HEX-graphs in Deng et al. (2014), propositional formulas in Xu et al. (2018), tractable circuits in Ahmed et al. (2022a), linear programs in Niepert et al. (2021), Prolog in Manhaeve et al. (2021), ASP in Yang et al., First Order Logic in Badreddine et al. (2022), etc. Because of this diversity, it is difficult to identify the shared concepts underlying most techniques introduced in the literature. It is also challenging to establish meaningful comparisons between techniques. In this

paper, we take a bird's eye view on informed classification and propose a unified formalism that encapsulates all representation languages for propositional knowledge. We build upon this formalism to study a family of neurosymbolic techniques that leverage probabilistic reasoning to integrate prior knowledge, a trend that has gained significant traction in the recent literature (Xu et al., 2018; Manhaeve et al., 2021; Ahmed et al., 2022a;b). We re-frame two existing neurosymbolic techniques: one that only impacts training (semantic regularization) and one that impacts both training and inference (semantic conditioning). We also define a new technique that only impacts the inference stage: **semantic conditioning at inference**. This is a particularly useful property in the era of **off-the-shelves** and **foundation** models (Bommasani et al., 2021), which are pre-trained on massive amounts of general data to then be applied in a multitude of heterogeneous downstream tasks.

**Contributions**    After preliminary notions (Section 2), we propose a unified formalism for **supervised multi-label classification informed by prior knowledge** (Section 3). Then, we introduce **semantic conditioning at inference** (Section 4). To the best of our knowledge, we are the first to define a neurosymbolic technique based on probabilistic reasoning which only impacts inference. We also analyze key properties and the computational complexity of neurosymbolic techniques based on probabilistic reasoning (Section 4.2). Finally, we evaluate the three techniques on several datasets, including large scale datasets whose sizes are rarely encountered in the neurosymbolic literature (Section 5).

## 2 PRELIMINARIES

In this section, we give the preliminary definitions needed for our unified formalism. We first give a general definition of representation languages for propositional knowledge, which we will use throughout the paper to express prior knowledge on multi-label classification tasks. Then, we define several probabilistic reasoning problems that will be the foundation of the neurosymbolic techniques we will study in the paper.

### 2.1 KNOWLEDGE REPRESENTATION

In its more abstract form, **knowledge** about a **world** tells us in what **states** this world can be observed. In this paper, we only consider propositional knowledge, where the states correspond to subsets of a discrete set of variables $\mathbf{Y}$ and knowledge tells us what combinations of variables can be observed in the world. The set of *possible* states is $\mathbb{B}^{\mathbf{Y}}$, where $\mathbb{B} := \{0, 1\}$. A state $\mathbf{y} \in \mathbb{B}^{\mathbf{Y}}$ can be seen as a subset of $\mathbf{Y}$ as well as an application that maps each variable to $\mathbb{B}$ (*i.e.*, for a variable $Y_i \in \mathbf{Y}$, $\mathbf{y}_i = 1$ is equivalent to $Y_i \in \mathbf{y}$). Knowledge defines a set of states that are considered *valid*. An *abstract* representation of this knowledge is a **boolean function** $f \in \mathbb{B}^{\mathbb{B}^{\mathbf{Y}}}$, which can be viewed either as a function that maps states in $\mathbb{B}^{\mathbf{Y}}$ to boolean values or as a subset of $\mathbb{B}^{\mathbf{Y}}$. However, in order to exploit this knowledge (*e.g.* reason, query, communicate, etc.), we need a *concrete* language to represent it.

**Definition 1** (Propositional language). A **propositional language** is a couple $\mathcal{F} := (\mathcal{T}, \jmath)$ such that for any discrete set of variables $\mathbf{Y}$:

- $\mathcal{T}(\mathbf{Y})$ is the set of admissible **theories** on $\mathbf{Y}$

- $\jmath(\mathbf{Y})$ determines which states on $\mathbf{Y}$ satisfy a theory $\kappa$:

$$\jmath(\mathbf{Y}) : \mathcal{T}(\mathbf{Y}) \to \mathbb{B}^{\mathbb{B}^{\mathbf{Y}}}$$

When the set of variables is clear from context, we simply note $T \in \mathcal{T}$ and $\jmath(T)$ in place of $T \in \mathcal{T}(\mathbf{Y})$ and $\jmath(\mathbf{Y})(T)$. We say that a propositional language $\mathcal{F}_2 := (\mathcal{T}_2, \jmath_2)$ is a **fragment** of a propositional language $\mathcal{F}_1 := (\mathcal{T}_1, \jmath_1)$, noted $\mathcal{F}_2 \subset \mathcal{F}_1$, iff for any discrete set of variables $\mathbf{Y}$: $\mathcal{T}_2(\mathbf{Y}) \subset \mathcal{T}_1(\mathbf{Y})$ and $\jmath_2(\mathbf{Y})$ is the restriction of $\jmath_1(\mathbf{Y})$ to $\mathcal{T}_2(\mathbf{Y})$. A state $\mathbf{y} \in \mathbb{B}^{\mathbf{Y}}$ **satisfies** a theory $\kappa \in \mathcal{T}$ iff $\mathbf{y} \in \jmath(\mathbf{Y})(T)$. We also say that $\kappa$ accepts $\mathbf{y}$. A theory $\kappa$ is **satisfiable** if it is satisfied by a state, *i.e.*, if $\jmath(\mathbf{Y})(\kappa) \neq \varnothing$. Two theories $\kappa_1$ and $\kappa_2$ are **equivalent** iff $\jmath(\kappa_1) = \jmath(\kappa_2)$.

Definition 1 covers many knowledge representation languages found in the literature. We illustrate below with propositional logic and detail several other propositional languages in Appendix A.

**Example 1.** *A theory in propositional logic is called a **propositional formula** and is formed inductively from variables and other formulas by using unary ($\neg$, which expresses negation) or binary ($\vee, \wedge$, which express disjunction and conjunction respectively) connectives. We note $\mathcal{T}_{PL}(\mathbf{Y})$ the set of formulas that can be formed in this way. The semantics of propositional logic can be inductively derived from the formula following the standard semantics of negation, conjunction and disjunction, i.e., a state $\mathbf{y}$ satisfies: a variable $Y_i \in \mathbf{Y}$ if $y_i = 1$, a formula $\neg\phi$ if $\phi$ is not satisfied by $\mathbf{y}$, a formula $\phi \vee \psi$ if $\mathbf{y}$ satisfies $\phi$ or $\psi$ and a formula $\phi \wedge \psi$ if $\mathbf{y}$ satisfies $\phi$ and $\psi$. For instance, $\kappa = Y_1 \wedge Y_2$ is satisfied by $\mathbf{y}$ iff $y_1 = y_2 = 1$. We refer to Russell & Norvig (2021) for more details on propositional logic.*

## 2.2 PROBABILISTIC REASONING

One challenge of neurosymbolic AI is to bridge the gap between the discrete nature of logic and the continuous nature of neural networks. Probabilistic reasoning can provide the interface between these two realms by allowing us to reason about uncertain facts.

A probability distribution on a set of **boolean variables** $\mathbf{Y}$ is an application $\mathcal{P} : \mathbb{B}^{\mathbf{Y}} \mapsto \mathbb{R}^+$ that maps each state $\mathbf{y}$ to a probability $\mathcal{P}(\mathbf{y})$, such that $\sum_{\mathbf{y} \in \mathbb{B}^{\mathbf{Y}}} \mathcal{P}(\mathbf{y}) = 1$. To define internal operations between distributions, like multiplication, we extend this definition to un-normalized distributions $\mathcal{E} : \mathbb{B}^{\mathbf{Y}} \mapsto \mathbb{R}^+$. The **null distribution** is the application that maps all states to 0. The **partition function** $\mathsf{Z} : \mathcal{E} \mapsto \sum_{\mathbf{y} \in \mathbb{B}^{\mathbf{Y}}} \mathcal{E}(\mathbf{y})$ maps each distribution to its sum, and we note $\overline{\mathcal{E}} := \frac{\mathcal{E}}{\mathsf{Z}(\mathcal{E})}$ the normalized distribution (when $\mathcal{E}$ is non-null). The **mode** of a distribution $\mathcal{E}$ is its most probable state, ie $\arg\max_{\mathbf{y} \in \mathbb{B}^{\mathbf{Y}}} \mathcal{E}(\mathbf{y})$.

A standard distribution is the **exponential probability distribution**, which is parameterized by a vector of logits $\mathbf{a} \in \mathbb{R}^k$, one for each variable in $\mathbf{Y}$, and corresponds to the joint distribution of independent Bernoulli variables $\mathcal{B}(p_i)_{1 \leqslant i \leqslant k}$ with $p_i = \mathsf{s}(a_i)$. The independent multi-label classification system (see Example 2) is build by following the probabilistic interpretation based on this distribution.

**Definition 2.** Given a vector $\mathbf{a} \in \mathbb{R}^k$, the **exponential distribution** is:

$$\mathcal{E}(\cdot|\mathbf{a}) : \mathbf{y} \mapsto \prod_{1 \leqslant i \leqslant k} e^{a_i \cdot y_i} \tag{1}$$

We will note $\mathcal{P}(\cdot|\mathbf{a}) = \overline{\mathcal{E}(\cdot|\mathbf{a})}$ the corresponding normalized probability distribution.

Typically, when belief about random variables is expressed through a probability distribution and new information is collected in the form of evidence (*i.e.*, a partial assignment of the variables), we are interested in two things: computing the probability of such evidence and updating our beliefs using Bayes' rules by conditioning the distribution on the evidence. Probabilistic reasoning allows us to perform the same operations with logical knowledge in place of evidence. Let's assume a probability distribution $\mathcal{P}$ on variables $\mathbf{Y} := \{Y_j\}_{1 \leqslant j \leqslant k}$ and a **satisfiable** theory $\kappa$ from a propositional language $\mathcal{F} := (\mathcal{T}, \jmath)$. Notice that the boolean function $\jmath(\kappa)$ is an un-normalized distribution on $\mathbf{Y}$.

**Definition 3.** The **probability** of $\kappa$ under $\mathcal{P}$ is:

$$\mathcal{P}(\kappa) := \mathsf{Z}(\mathcal{P} \cdot \jmath(\kappa)) = \sum_{\mathbf{y} \in \mathbb{B}^{\mathbf{Y}}} \mathcal{P}(\mathbf{y}) \cdot \jmath(\kappa)(\mathbf{y}) \tag{2}$$

The distribution $\mathcal{P}$ **conditioned on** $\kappa$, noted $\mathcal{P}(\cdot|\kappa)$, is:

$$\mathcal{P}(\cdot|\kappa) := \overline{\mathcal{P} \cdot \jmath(\kappa)} \tag{3}$$

Since $\mathcal{P}(\cdot|\mathbf{a})$ is strictly positive (for all $\mathbf{a}$), if $\kappa$ is satisfiable then its probability under $\mathcal{P}(\cdot|\mathbf{a})$ is also strictly positive. We note:

$$\mathcal{P}(\kappa|\mathbf{a}) := \mathsf{Z}(\mathcal{P}(\cdot|\mathbf{a}) \cdot \jmath(\kappa)) \qquad\qquad \mathcal{P}(\cdot|\mathbf{a}, \kappa) := \frac{\mathcal{P}(\cdot|\mathbf{a}) \cdot \jmath(\kappa)}{\mathcal{P}(\kappa|\mathbf{a})}$$

Computing $\mathcal{P}(\kappa|\mathbf{a})$ is a **counting** problem called **Probabilistic Query Estimation** (PQE). Computing the mode of $\mathcal{P}(\cdot|\mathbf{a}, \kappa)$ is an **optimization** problem called **Most Probable Explanation** (MPE). Solving these probabilistic reasoning problems is at the core of many neurosymbolic techniques, as shown in Section 4.

## 3 INFORMED SUPERVISED CLASSIFICATION

In this section, we introduce a formalism for informed supervised classification. We first detail our computational framework for neural multi-label classification systems, which will serve as our basis for neurosymbolic techniques, then we define what we call informed classification tasks.

### 3.1 NEURAL MULTI-LABEL CLASSIFICATION

In supervised machine learning, the objective is to learn a relationship between an **input domain** $\mathcal{X}$ and an **output domain** $\mathcal{Y}$ from a labeled dataset $\mathsf{D} := (x^i, \mathbf{y}^i)_{1 \leqslant i \leqslant d} \in (\mathcal{X} \times \mathcal{Y})^d$.

Deep learning systems usually adopts a *functional framework* to tackle supervised learning tasks. A functional relation $f : \mathcal{X} \mapsto \mathcal{Y}$ is assumed and a neural network (*i.e.*, a parametric and differentiable computational graph) $\mathsf{M}$ is designed to model this relation based on assumed properties of $f$. To learn the parameters of the neural network a differentiable cost function $\mathsf{L}$ measuring the distance between predictions and labels is chosen, backpropagation computes the gradient of the loss with respect to each parameter and gradient descent is used to minimize the empirical error. Inference is done by processing the inputs through the neural network.

However, when the output domain $\mathcal{Y}$ is (at least partly) **discrete**, a differentiable distance cannot be defined directly on $\mathcal{Y}$ and such a framework cannot be applied strictly. This is especially relevant for classification tasks. Classification tasks are usually **categorical**: the output domain consists of a finite set of variables. In **multi-label classification** tasks however, elements in the output domain $\mathcal{Y}$ are subsets of a finite set of classes $\mathbf{Y}$. Usually called **labelsets**, we call them **states** (see Section 2.1) and note $\mathcal{Y} = \mathbb{B}^{\mathbf{Y}}$. Interestingly, categorical classification can be seen as a specific instance of multi-label classification informed with prior knowledge (see Example 4).

Hence, we adopt a slightly modified framework, called *pseudo-functional*, where a third module $\mathsf{I}$ (besides $\mathsf{M}$ and $\mathsf{L}$), called the inference module, has to be defined to bridge the gap between the continuous nature of the neural network (needed for gradient descent) and the discrete nature of the output space. This third module, although essential, is not often explicitly described. An illustration can be found on Figure 1.

**Definition 4.** A **neural classification system** for multi-label classification is the given of :

- a **parametric differentiable** (*i.e.*, neural) module $\mathsf{M}$, called the **model**, which takes as inputs $x \in \mathbb{R}^d$, parameters $\theta \in \Theta$ and outputs $\mathsf{M}(x, \theta) := \mathsf{M}_\theta(x) := \mathbf{a} \in \mathbb{R}^k$, called **pre-activation scores** or **logits**.

- a **non-parametric differentiable** module $\mathsf{L}$, called the **loss** module, which takes $\mathbf{a} \in \mathbb{R}^k$ and $\mathbf{y} \in \{0, 1\}^k$ as inputs and outputs a scalar.

- a **non-parametric** module $\mathsf{I}$, called the **inference** module, which takes $\mathbf{a} \in \mathbb{R}^k$ as input and outputs a prediction $\hat{\mathbf{y}} \in \{0, 1\}^k$.

*Remark* 1. For lighter notations, we note $\mathbf{a} \in \mathbb{R}^k$ as a simpler notation for $\mathbf{a} \in \mathbb{R}^{\mathbf{Y}}$ assuming $\mathbf{Y} := \{Y_j\}_{1 \leqslant j \leqslant k}$.

A common approach to design a neural classification system is to build upon a natural probabilistic interpretation. Logits produced by the neural network are seen as parameters of a conditional probability distribution of the output given the input $\mathcal{P}(\cdot | \mathsf{M}_\theta(x))$, the loss module computes the cross-entropy of that distribution with a ground truth label, and the inference module computes the most probable output given the learned distribution.

When no prior knowledge is available about the set of classes (uninformed case), a standard hypothesis is to assume independent output variables. We illustrate below how this translates in a specific neural classification system.

**Example 2.** *For **independent multi-label** classification, we apply a sigmoid layer on logits to turn them into probability scores. The loss is the binary cross-entropy (BCE) between probability scores and labels, and a variable is predicted to be true if its probability is above $0.5$ (or equivalently its*

*logit is above* 0*). This results in the following modules:*

$$\mathsf{L}_{imc}(\mathbf{a}, \mathbf{y}) := \mathtt{BCE}(\mathsf{s}(\mathbf{a}), \mathbf{y})$$
$$= -\sum_j y_j . \log(\mathsf{s}(a_j)) + (1 - y_j) . \log(1 - \mathsf{s}(a_j)) \tag{4}$$

$$\mathsf{l}_{imc}(\mathbf{a}) := \mathbb{1}[\mathbf{a} \geqslant 0] \tag{5}$$

*where* $\mathsf{s}(a_i) = \frac{e^{a_i}}{1 + e^{a_i}}$ *is the sigmoid function and* $\mathbb{1}[z] := \begin{cases} 1 & \textit{if z true} \\ 0 & \textit{otherwise} \end{cases}$ *the indicator function.*

## 3.2 TASK

A task of supervised multi-label classification is **informed** when it comes attached with prior knowledge, expressed as a theory $\kappa$ in a propositional language $\mathcal{F} := (\mathcal{T}, \jmath)$, specifying which states in the output space are **semantically valid**.

A supervised dataset $\mathsf{D} := (x^i, \mathbf{y}^i)_{1 \leqslant i \leqslant d} \in (\mathcal{X} \times \mathcal{Y})^d$ is **consistent** with prior knowledge $\kappa$ if all labels satisfy $\kappa$ (*i.e.*, $\forall 1 \leqslant i \leqslant n, \mathbf{y}^i \models \kappa$). In this paper we will work under the hypothesis that both training and test datasets are consistent. However, some techniques allow for a relaxation of this assumption, enabling the use of inconsistent datasets.

## 4 TECHNIQUES

When prior knowledge is available about a classification task, it seems only natural to improve our neural classification system by integrating this knowledge into its design. We give below two examples of informed classification tasks and how the loss and inference modules can be adapted to embed prior knowledge.

**Example 3.** *Categorical classification arises when one and only one output variable is true for a given input sample (e.g. mapping an image to a single digit in $[\![0, 9]\!]$ for MNIST). These constraints can easily be enforced by the following propositional formula:*

$$\kappa_{\odot_k} := \left( \bigvee_{1 \leqslant j \leqslant k} Y_j \right) \wedge \left( \bigwedge_{1 \leqslant j < l \leqslant k} (\neg Y_j \vee \neg Y_l) \right) \tag{6}$$

*where the first part ensures that at least one variable is true and the second part prevents two variables to be true simultaneously. For categorical classification, the sigmoid layer is replaced by a softmax layer and the variable with the maximum score is predicted, which leads to the following modules:*

$$\mathsf{L}_{\odot_k}(\mathbf{a}, \mathbf{y}) := \mathtt{CE}(\mathsf{s}(\mathbf{a}), \mathbf{y}) = -\log(\langle \sigma(\mathbf{a}), \odot_k(j) \rangle) \tag{7}$$
$$\mathsf{l}_{\odot_k}(\mathbf{a}) := \odot_k(\arg\max(\mathbf{a})) \tag{8}$$

*where* CE *is the cross-entropy,* $\sigma(\mathbf{a}) = (\frac{e^{a_j}}{\sum_l e^{a_l}})_{1 \leqslant j \leqslant k}$ *and* $\odot_k$ *gives the one-hot encoding (starting at 1) of* $j \in [\![1, k]\!]$, *e.g.* $\odot_4(2) = (0, 1, 0, 0)$.

**Example 4.** *Hierarchical classification on a set of variables* $\{Y_j\}_{1 \leqslant j \leqslant k}$ *is usually formulated with a directed acyclic graph* $G = (Y, E_h)$ *where the nodes are the variables and the edges* $E_h$ *express subsumption between those variables (e.g. a dog is an animal). This formalism can even be enriched with exclusion edges* $H = (Y, E_h, E_e)$ *(e.g. an input cannot be both a dog and a cat), like in HEX-graphs (Deng et al., 2014). There again, the translation to propositional logic is straightforward:*

$$\kappa_H := \left( \bigwedge_{(i,j) \in E_h} Y_i \vee \neg Y_j \right) \wedge \left( \bigwedge_{(i,j) \in E_e} (\neg Y_i \vee \neg Y_j) \right) \tag{9}$$

*where the first part ensures that a son node cannot be true if its father node is not and the second part prevents two mutually exclusive nodes to be true simultaneously. Many techniques have been proposed to integrate hierarchical knowledge in a neural classification system. For instance, (Muller & Smith, 2020) introduces a hierarchical loss to penalize more errors on higher classes of the hierarchy, (Giunchiglia & Lukasiewicz, 2020) refines the logits based on the hierarchy while (Deng et al., 2014) replaces the exponential distribution by a Conditional Random Field that integrates the hierarchical knowledge.*

Beyond categorical and hierarchical classification, propositional knowledge can be used to define very diverse output spaces: *e.g.* Sudoku solutions (Augustine et al., 2022), simple paths in a graph (Xu et al., 2018; Ahmed et al., 2022a), preference rankings (Xu et al., 2018), matchings in a graph (Pogančić et al., 2019; Ahmed et al., 2022b), etc.

Therefore, the purpose of a neurosymbolic technique is to automatically derive appropriate loss and inference modules from prior knowledge, **generalizing the work made on categorical and hierarchical cases to arbitrary structures**. We formalize this process with the definition below and illustrate it on Figure 1.

**Definition 5** (Neurosymbolic technique). A **neurosymbolic technique** for a propositional language $\mathcal{F} := (\mathcal{T}, \jmath)$ is $\mathfrak{T} := (\mathfrak{L}, \mathfrak{I})$ such that for any finite set of variables $\mathbf{Y}$ and theory $\kappa \in \mathcal{T}(\mathbf{Y})$:

$$\mathfrak{L}(\mathbf{Y}, \kappa) := \mathsf{L} : \mathbb{R}^k \times \mathcal{Y} \mapsto \mathbb{R}^+$$

$$\mathfrak{I}(\mathbf{Y}, \kappa) := \mathsf{I} : \mathbb{R}^k \mapsto \mathcal{Y}$$

*Remark 2.* A neurosymbolic technique *in general* is not an algorithm, but only a mathematical construct. Therefore, it gives no insight *a priori* into how the modules should be implemented.

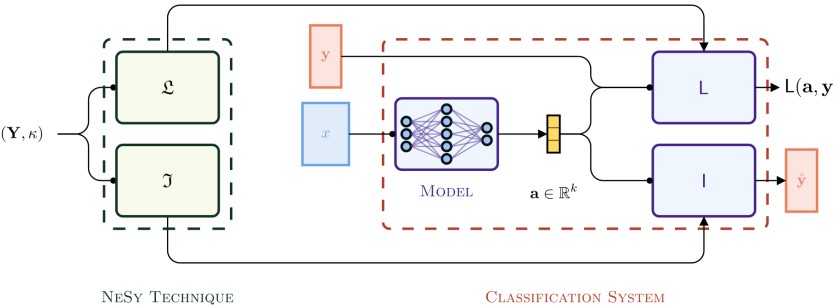

Figure 1: Illustration of a neurosymbolic technique $\mathfrak{T} := (\mathfrak{L}, \mathfrak{I})$: it takes prior knowledge $\kappa$ as input and outputs the loss $\mathsf{L}$ and inference $\mathsf{I}$ modules of a neural classification system.

**Fuzzy regularization** One of the first family of neurosymbolic techniques introduced in the literature was based on fuzzy regularization (Diligenti et al., 2017; Giannini et al., 2023; Badreddine et al., 2022): the fuzzy valuation of prior knowledge $\kappa$ (expressed in propositional logic) based on output scores $\mathsf{M}_\theta(x)$ is added to the standard negative log-likelihood of the labels to steer the model towards *valid* states.

### 4.1 PROBABILISTIC TECHNIQUES

Besides fuzzy logics, another paradigm that gained traction in the recent years as a basis for designing neurosymbolic techniques is probabilistic reasoning. We define below three neurosymbolic techniques based on probabilistic reasoning, including our new technique called **semantic conditioning at inference**, and relate each technique to the existing neurosymbolic literature.

**Semantic regularization** Similar to fuzzy regularization, semantic regularization uses the probability of the prior knowledge based on output scores $\mathsf{M}_\theta(x)$ (see Section 2.2) as a regularization term. It was introduced for propositional knowledge as the semantic loss in (Xu et al., 2018).

**Definition 6. Semantic regularization** (with coefficient $\lambda > 0$) for a propositional language $\mathcal{F} := (\mathcal{T}, \jmath)$ is $\mathfrak{T}_r^\lambda := (\mathfrak{L}_r^\lambda, \mathfrak{I}_r^\lambda)$ such that for any finite set of variables $\mathbf{Y}$ and theory $\kappa \in \mathcal{T}(\mathbf{Y})$:

$$\mathfrak{L}_r^\lambda(\mathbf{Y}, \kappa) : (\mathbf{a}, \mathbf{y}) \to -\log(\mathcal{P}(\mathbf{y}|\mathbf{a})) - \lambda.\log(\mathcal{P}(\kappa|\mathbf{a})) \tag{10}$$

$$\mathfrak{I}_r^\lambda(\mathbf{Y}, \kappa) : \mathbf{a} \to \underset{\mathbf{y} \in \mathbb{B}^\mathbf{Y}}{\arg\max} \mathcal{P}(\mathbf{y}|\mathbf{a}) \tag{11}$$

**Semantic conditioning** Following the probabilistic interpretation mentioned in Section 3.1, a natural way to integrate prior knowledge $\kappa$ into a neural classification system is to condition the dis-

tribution $\mathcal{P}(\cdot|\mathsf{M}(x,\theta))$ on $\kappa$. This conditioning affects the loss and inference modules, both underpinned by the conditional distribution. It was first introduced in (Deng et al., 2014) for Hierarchical-Exclusion (HEX) graphs constraints. Semantic probabilistic layers (Ahmed et al., 2022a) can be used to implement semantic conditioning on tractable circuits. NeurASP (Yang et al.) defines semantic conditioning on a predicate extension of ASP programs. An approached method for semantic conditioning on linear programs is proposed in (Niepert et al., 2021).

**Definition 7. Semantic conditioning** for a propositional language $\mathcal{F} := (\mathcal{T}, \jmath)$ is $\mathfrak{T}_{sc} := (\mathfrak{L}_{sc}, \mathfrak{I}_{sc})$ such that for any finite set of variables $\mathbf{Y}$ and theory $\kappa \in \mathcal{T}(\mathbf{Y})$:

$$\mathfrak{L}_{sc}(\mathbf{Y}, \kappa) : (\mathbf{a}, \mathbf{y}) \to -\log(\mathcal{P}(\mathbf{y}|\mathbf{a}, \kappa)) \tag{12}$$

$$\mathfrak{I}_{sc}(\mathbf{Y}, \kappa) : \mathbf{a} \to \underset{\mathbf{y} \in \mathbb{B}^{\mathbf{Y}}}{\arg\max} \mathcal{P}(\mathbf{y}|\mathbf{a}, \kappa) \tag{13}$$

**Semantic conditioning at inference** Finally, we introduce a new neurosymbolic technique, called **semantic conditioning at inference**, which is derived from semantic conditioning but only applies conditioning in the inference module (*i.e.*, infers the most probable state that satisfies prior knowledge) while retaining the standard negative log-likelihood loss.

**Definition 8. Semantic conditioning at inference** for a propositional language $\mathcal{F} := (\mathcal{T}, \jmath)$ is $\mathfrak{T}_{sci} := (\mathfrak{L}_{sci}, \mathfrak{I}_{sci})$ such that for any finite set of variables $\mathbf{Y}$ and theory $\kappa \in \mathcal{T}(\mathbf{Y})$:

$$\mathfrak{L}_{sci}(\mathbf{Y}, \kappa) : (\mathbf{a}, \mathbf{y}) \to -\log(\mathcal{P}(\mathbf{y}|\mathbf{a})) \tag{14}$$

$$\mathfrak{I}_{sci}(\mathbf{Y}, \kappa) = \mathfrak{I}_{sc}(\mathbf{Y}, \kappa) \tag{15}$$

### 4.2 PROPERTIES

As shown above, Definition 5 encapsulates very diverse neurosymbolic techniques. However, beyond this unified view, we can analyze specific properties of neurosymbolic techniques that are critical to their deployment. Formal definitions and proofs can be found in Appendix B.

**Syntactic invariance** A neurosymbolic technique is **invariant to syntax** when equivalent formulas produce identical loss and inference modules when fed to $\mathfrak{L}$ and $\mathfrak{I}$. Interestingly, since $\mathfrak{L}(\mathbf{Y}, \kappa)$ and $\mathfrak{I}(\mathbf{Y}, \kappa)$ essentially depend on the boolean function represented by $\kappa$ and not on the syntax $\kappa$, it allows to generalize the technique defined for a particular propositional language to other languages. Because probabilistic reasoning essentially depends on the boolean function represented by the theory $\kappa$ and not its syntax, techniques that rely on probabilistic reasoning (*e.g.* see Definitions 6, 7 and 8) are *naturally* invariant to syntax. The case of semantic conditioning, which was introduced several times in the neurosymbolic literature using various knowledge representation languages, illustrates the importance of this property and the utility of this unified view of informed classification. On the contrary, because fuzzy regularization is based on the syntax of propositional formulas, it cannot be easily generalized to other representation languages. Besides, this sensitivity to syntax implies that two equivalent propositional formulas (i.e., representing the same prior knowledge) may produce different regularization terms, which could lead to performance disparities that are challenging to elucidate to end users. Finally, even when the mathematical definition of a technique can be easily generalized, this does not mean that its implementation and computational complexity is equivalent regardless of the propositional language.

**Consistency** A neurosymbolic technique is **consistent** (defined in (Ahmed et al., 2022a)) when the inference module can only produce outputs that satisfy the prior knowledge. By definition, techniques that only impact the loss module (*e.g.* techniques based on regularization terms) cannot verify this property, whereas the inference module of semantic conditioning guarantees consistency.

Besides retaining syntactic invariance and consistency from semantic conditioning, semantic conditioning at inference has other useful properties that make it a suitable choice compared to semantic conditioning and regularization. First, we show in Section 4.3 that it is more tractable computationally. Second, integrating prior knowledge only at inference time offers more flexibility than integration during training. For instance, it can be used if prior knowledge is unavailable at training time (for instance Giunchiglia et al. (2023) provides prior knowledge to an existing task of object detection) or susceptible to evolve. This is a particularly useful property in the era of **off-the-shelves**

and **foundation** models (Bommasani et al., 2021), which are pre-trained on massive amounts of general data to then be applied in a multitude of heterogeneous downstream tasks, since task specific prior knowledge can not be integrated during most of the training process.

### 4.3 A LOOK ON COMPLEXITY

As mentioned in Section 2.2, all three neurosymbolic techniques defined in Section 4 internally rely on solving MPE and PQE problems. Unfortunately, MPE and PQE are NP-hard and #P-hard respectively for most propositional languages commonly used to represent knowledge (*e.g.* propositional logic, boolean circuits, linear programs, ASP, etc.). This implies that scaling probabilistic neurosymbolic techniques to large classification tasks (*i.e.*, tasks with a large number of variables) on arbitrary prior knowledge requires an exponential amount of computing resources (unless P = NP) and is therefore not realistic. Hence, it is critical for any technique to identify (as much as possible) its domain of tractability, which is rarely done in the neurosymbolic literature.

Hopefully, there are fragments of propositional languages for which MPE and PQE are tractable. Boolean circuits in Decomposable Negational Normal Form (DNNF) can solve MPE problems in linear time (in the size of the circuit) and deterministic-DNNF (dDNNF) can additionally solve PQE problems in linear time. An approach that has become predominant in the literature is **knowledge compilation** (Darwiche & Marquis), which consists in translating a theory from an initial propositional language (*e.g.* CNF) into a target propositional language that can solve reasoning problems efficiently (*i.e.*, in a time polynomial in the size of the compiled formula).

Finally, counting problems are known to be much harder in general than optimization problems (Toda, 1991). For instance, MPE can be solved in polynomial time for formulas representing matching constraints (by reduction to finding a maximum weight-sum matching (Edmonds)), while PQE is still #P-hard (Amarilli & Monet). As semantic conditioning at inference only relies on solving MPE for its inference module, compared to semantic conditioning and semantic regularization which both rely on solving PQE to compute their loss module, this implies that semantic conditioning at inference remains tractable for a larger class of tasks than semantic conditioning and semantic regularization.

## 5 EXPERIMENTS ON LARGE SCALE DATA

In this section, we evaluate empirically the impact of neurosymbolic techniques on four informed classification tasks: a categorical task, two hierarchical tasks and a simple path prediction task.

### 5.1 A NEW MULTI-SCALE EVALUATION

Most papers in the field evaluate the benefits of their neurosymbolic technique on a single neural network architecture. Although informative, such a methodology paints a very limited picture of the benefits of the technique and leaves many questions unanswered. In particular, it does not allow to estimate how these benefits evolve when resources given to the system (*e.g.* network scale, dataset size, training time, etc.) increase.

To overcome those limitations, we selected for each task a single architectural design that can be scaled to various sizes (e.g. DenseNets (Huang et al., 2017)) and compared the performance of the three neurosymbolic techniques against an uninformed baseline (independent multi-label classification) across network scales. We report the **exact accuracy** (Ahmed et al., 2022a) (called coherent accuracy in Deng et al. (2014)), *i.e.*, the share of instances which are well classified on all labels, as our evaluation metric. More details on the methodology and the experimental setup (number of epochs, hyperparameters, etc.) are given in Appendix C.

### 5.2 DATASETS

We evaluate the three techniques on four different tasks: a categorical task based on MNIST dataset (LeCun et al., 1998), two hierarchical tasks based on Cifar-100 (Krizhevsky, 2009) and ImageNet (Russakovsky et al., 2015) datasets and an acyclic simple path prediction task based on Warcraft Shortest Path (WSP) dataset (Pogančić et al., 2019). For WSP tasks featured in the neurosymbolic

literature (Yang et al.; Niepert et al., 2021; Ahmed et al., 2022a), MPE and PQE are intractable and cannot be scaled to large grids. We modify the WSP dataset to make the graph acyclic, and develop an algorithm to compile acyclic simple path constraints into Ordered Binary Decision Diagrams (a fragment of dDNNF). Therefore, for each type of tasks tackled in our experiments, prior knowledge can be compiled into a polysize dDNNF, allowing to solve MPE and PQE tractably and scale properly with larger of set of variables. See Appendix C.1 for more details.

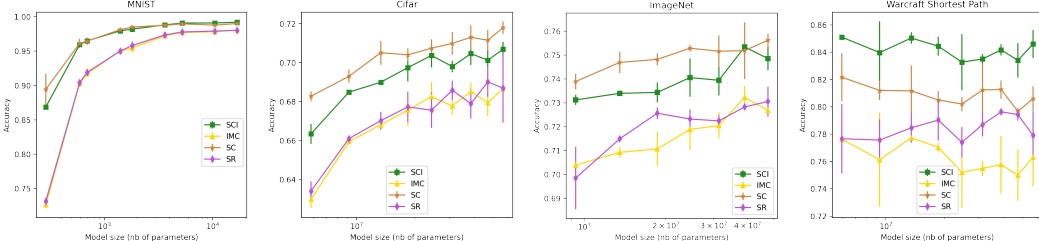

Figure 2: From left to right: each graph plots the exact accuracy on MNIST, Cifar, Imagenet and Warcraft Shortest Path, for all four techniques, against the size of the network. Errorbars represent the standard variation after aggregation on several seeds.

### 5.3 RESULTS AND ANALYSIS

The results of our experiments are displayed on Figure 2, a graphical representation has been chosen over a tabular one to highlight how accuracy curves evolve as the network scales.

**Observation 1. Semantic conditioning and semantic conditioning at inference outperform semantic regularization and independent multi-label classification** across tasks and model scales.

**Observation 2.** Except for the larger networks on Warcraft Shortest Path, **semantic regularization brings little benefits** in terms of accuracy compared to independent multi-label classification.

**Observation 3.** On MNIST, Cifar and ImageNet, **semantic conditioning at inference retains most of the performance gains** (about 75%) **of semantic conditioning**, despite only integrating knowledge during inference. It even **outperforms semantic conditioning on Warcraft Shortest Path**.

We expected semantic conditioning to outperform semantic conditioning at inference, since prior knowledge is integrated in the loss module as well as in the inference module, but experiments on Warcraft Shortest Path shows a different picture.

**Observation 4.** Accuracy gains of semantic conditioning at inference **tend to decrease** and converge towards a significantly positive value as the accuracy of the neural network increases.

Besides, on MNIST, Cifar and ImageNet, marginal accuracy gains with respect to the network scale are decreasing. In other terms, reaching a 1% accuracy improvement by scaling the network requires much more additional parameters larger networks than for smaller networks. Accuracy gains obtained from the integration of prior knowledge work in a similar fashion: they decrease with the size of the network and converge towards a significantly positive value, meaning that these techniques can improve performances on networks of all sizes. On Warcraft Shortest Path, as the accuracy slightly decreases with network scale, the accuracy gains of semantic conditioning increase with network scale.

The difference of behavior between MNIST, Cifar and ImageNet *vs.* Warcraft Shortest Path could be due to many factors: the nature of the prior knowledge (categorical and hierarchical *vs.* simple path), the nature the images (*real life* images *vs.* synthetic images), etc. More experiences with diverse informed classification tasks are needed to elucidate this point.

Anyhow, our experiments strongly suggests that, in the supervised setting, **the most critical module for knowledge integration is the inference module**.

## 6 RELATED WORK

In this paper, we restricted our formalism to supervised classification tasks informed with propositional prior knowledge. However, many techniques in the literature work with prior knowledge expressed in a higher order language or solve classification tasks where full supervision is lacking.

**Predicate languages** Propositional languages use propositional variables to represent atomic facts, which constitute the smallest unit of discourse to represent the world. Predicate languages decompose atomic facts into a more fine grained representation, then leverage this compositional representation through quantification to provide a more expressive language. Predicate languages are often used in the neurosymbolic literature (*e.g.* First Order Logic in (Badreddine et al., 2022), Prolog in (Manhaeve et al., 2021), ASP in (Yang et al.)) to represent compositional knowledge about the input space or impose a structural bias on the neural architecture. However, probabilistic neurosymbolic techniques systematically rely on grounding to perform probabilistic reasoning, which limits the impact of predicate languages on the computational side.

**Supervision settings** In real world applications, labeling large amounts of data is difficult, expensive and slow, especially for multi-label classification tasks featuring many classes (Deng et al.). Therefore, much work has been done to formalize and exploit cheaper supervision settings where input samples are not fully labeled. In the **semi-supervised** setting (Seeger, 2000; Grandvalet & Bengio, 2004), only a fraction of input samples are fully labeled while the rest is unlabeled. Closely related is the **partial-labels** setting (Durand et al.), where only a subset of the classes are labeled for each input sample. Partial labels can typically be found when prior knowledge represents a functional dependency between a set of latent variables and a set of observed variables, like in the MNIST-Add task (Manhaeve et al., 2021; Badreddine et al., 2022; Maene & De Raedt; van Krieken et al.), which aim is to learn a latent representation of hand-written digits from observing only their sum. Some neurosymbolic techniques have been specifically designed for these supervision settings (Xu et al., 2018; Ahmed et al., 2022b).

## 7 CONCLUSION

In this paper, we introduce a **formalism for supervised classification informed by prior knowledge**, define a new neurosymbolic technique called **semantic conditioning at inference** which integrates this prior knowledge during inference. To the best of our knowledge, this is the first neurosymbolic technique based on probabilistic reasoning which only impacts inference. We evaluate our technique alongside two existing probabilistic techniques on several large datasets and across neural network scales. We show experimentally that **semantic conditioning at inference can improve the performances of a neural classification system on large datasets and on networks of all sizes**. Besides, we demonstrate that semantic conditioning at inference **preserves key properties** (*i.e.*, syntactic invariance and consistency) and **remains competitive** with semantic conditioning while **only working at inference**, making it **more flexible and tractable**.

Future directions for our work may include, amongst others, reproducing our experiments on more datasets, investigating the semi-supervised and partial-labels settings. Finally, we assume throughout the paper that the knowledge is known *a priori*, which is often not the case in practice. Discovering the *structure* of the task at hand and training the model simultaneously is an important field of research.

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

## A  KNOWLEDGE REPRESENTATION

We detail in this section several other knowledge representation languages. For each language, we first define its syntax then detail its semantics. Other representation languages of boolean functions include decision lists (Rivest), decision trees (Krzywinski & Altman; Quinlan), random forests (Breiman), boosted trees (Freund & Schapire) or binarized neural networks (Hubara et al.).

### A.1  CIRCUITS

Boolean circuits (Darwiche, 2009) $\mathcal{F}_C := (\mathcal{C}, s_C)$ is a representation language that has gained a lot of traction in recent years because some of its fragments provide tractable algorithms of many reasoning tasks.

A **boolean circuit** $C \in \mathcal{C}(\mathbf{Y})$ on variables $\mathbf{Y}$ is a couple $C := (G, \varsigma)$ where:

- $G = (N, W)$ is a directed acyclic graph
- vertices in $N$ are called **nodes** and edges in $W$ are called **wires**
- $G$ has a single root $r$ (*i.e.*, a node without parents)
- $\varsigma : N \to \mathbf{Y} \cup \{\top, \bot, \neg, \wedge, \vee\}$ such that:
  - $\varsigma(n) \in \mathbf{Y} \cup \{\top, \bot\}$ iff $n$ is a leaf node
  - $\varsigma(n) = \neg$ iff $n$ has exactly one child
  - $\varsigma(n) \in \{\wedge, \vee\}$ iff $n$ has at least two children

The set of children of a node $n \in N$ is noted $ch(n)$. The set of variables of a circuit is noted $var(C)$. Given a node $n \in N$, we note $C^n$ the circuit obtained by keeping all nodes that are descendants of $n$ in $G$. We sometimes note $var(n)$ for $var(C^n)$.

Let's assume a state $\mathbf{y} \in \mathbb{B}^{\mathbf{Y}}$ and a circuit $C := (G, \varsigma) \in \mathcal{C}(\mathbf{Y})$ of root $r$. To know if $\mathbf{y}$ satisfies the circuit (*i.e.*, $s_C(C)(\mathbf{y}) = 1$, we evaluate the circuit bottom up, mapping each node $n$ to 0 or 1. First, leaf nodes $n$ are mapped to 1 if $\varsigma(n) = \top$, to 0 if $\varsigma(n) = \bot$ and to $\mathbf{y}(\varsigma(n))$ if $\varsigma(n) \in \mathbf{Y}$. Then, for any internal node $n$, it is valued 1 if $\varsigma(n) = \neg$ and its child is valued 1, or if $\varsigma(n) = \vee$ and one of its children is valued 1 or if $\varsigma(n) = \wedge$ and all its children are valued 1. Otherwise it is valued 0. The state $\mathbf{y}$ satisfies the circuit if the root is valued at 1.

A circuit is in **Negational Normal Form** (NNF) if all negation nodes have variables as children, *i.e.*, :
$$\forall (u, v) \in W, \varsigma(u) = \neg \implies \varsigma(v) \in \mathbf{Y}$$
(*i.e.*, $\forall (u, v) \in W, \varsigma(u) = \neg \implies \varsigma(v) \in \mathbf{Y}$).

A conjunction node $u$ (*i.e.*, $\varsigma(u) = \wedge$) is **decomposable** if the sub-circuits rooted in each of its children do not share variables, *i.e.*, :
$$\forall (u, v), (u, w) \in W, var(v) \cup var(w) = \varnothing$$

A circuit is in **Decomposable Negational Normal Form** (DNNF) if it is NNF and all of its conjunction nodes are decomposable.

A disjunction node $u$ (*i.e.*, $\varsigma(u) = \vee$) is **deterministic** if the sub-circuits rooted in each of its children do not share satisfying assignments, *i.e.*, :
$$\forall (u, v), (u, w) \in W, s_C(C^v) \cup s_C(C^w) = \varnothing$$

A circuit is in **Deterministic Decomposable Negational Normal Form** (dDNNF) if it is DNNF and all of its disjunction nodes are deterministic.

Besides, any propositional formula can be translated in linear time into an equivalent boolean circuit by reading the formula in the standard priority order. Therefore, usual fragments of propositional logic (*e.g.* CNF, DNF, etc.) translate into fragments of boolean circuits.

Recently, (Amarilli et al., 2024) showed that decision diagrams (*e.g.* Ordered Binary Decision Diagrams) also correspond to fragments of boolean circuits.

A map of fragments of boolean circuits is represented on Figure 3.

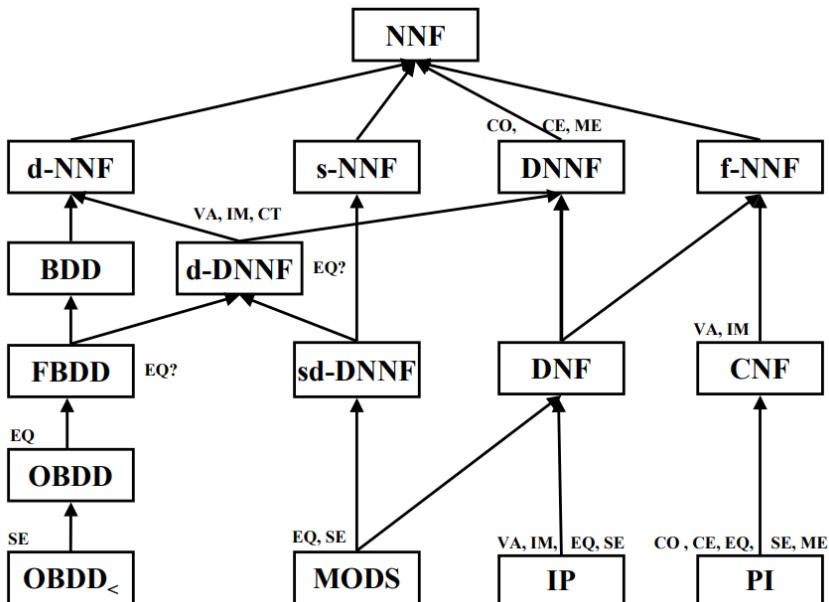

Figure 3: Fragments of boolean circuits in Darwiche (2009): an edge $\mathcal{F}_2 \longrightarrow \mathcal{F}_1$ means that $\mathcal{F}_2$ is a fragment of $\mathcal{F}_1$.

## A.2 ANSWER SET PROGRAMMING

Answer Set Programming (ASP) (Faber, 2020) $\mathcal{F}_{ASP} := (\mathcal{T}_{ASP}, \jmath_{ASP})$ is one of the simplest examples of non-monotonic logics, which enable concise representations of complex knowledge at the cost of monotonicity.

A theory $\Pi \in \mathcal{T}_{ASP}(\mathbf{Y})$ in ASP is composed of a set of rules and is called a **program**. A **rule** $r \in \mathcal{R}(\mathbf{Y})$ is formed from the grammar:

$$r :== a.|h \leftarrow b \quad h :== a|\bot \quad b :== l|b, l \quad l :== a|\text{not } a$$

where $\{a \in \mathbf{Y}, \leftarrow, \text{not}\}$ are terminal symbols.

$h$ and $b$ are respectively called the **head** and the **body** of the rule.

**Example 5.** $r := Y_1 \leftarrow Y_2, \text{not } Y_3$ *is a rule on variables* $\mathbf{Y} = \{Y_1, Y_2, Y_3\}$.

The **reduct** of a program $\Pi \in \mathcal{T}_{ASP}(\mathbf{Y})$ relative to a state $\mathbf{y} \in \mathbb{B}^{\mathbf{Y}}$ is the program $\Pi^{\mathbf{y}}$ :

$$\Pi^{\mathbf{y}} = \{Y_{i_0} \leftarrow Y_{i_1}, ..., Y_{i_l}.|r \in \Pi, r : Y_{i_0} \leftarrow Y_{i_1}, ..., Y_{i_l}, \text{not } Y_{i_{l+1}}, ..., \text{not } Y_{i_m}., \forall j \in \{i_{l+1}, ..., i_m\}, y_j = 0\}$$

To get $\Pi^{\mathbf{y}}$, we first eliminate all rules in $\Pi$ such that $\mathbf{y}$ does not satisfy the negative part of the body, then for remaining rules, we delete the negative part of the body and add them to $\Pi^{\mathbf{y}}$.

We say that a state $\mathbf{y} \in \mathbb{B}^{\mathbf{Y}}$ **fires** a rule $r : Y_{i_0} \leftarrow Y_{i_1}, ..., Y_{i_l}, \text{not } Y_{i_{l+1}}, ..., \text{not } Y_{i_m}.$ iff $\mathbf{y}$ satisfies $\phi_r$ (in the sense of propositional logic) where:

$$\phi_r := \neg Y_{i_1} \vee ... \vee \neg Y_{i_l} \vee Y_{i_{l+1}} \vee Y_{i_m} \vee \neg Y_{i_0}$$

Finally, $\jmath_{ASP}$ follows the **answer set semantics**: a state $\mathbf{y} \in \mathbb{B}^{\mathbf{Y}}$ in an **answer set** for a program $\Pi$, noted $\jmath_{ASP}(\Pi)(\mathbf{y}) = 1$, iff it is the smaller state (in terms of inclusion) to fire all rules of $\Pi^{\mathbf{y}}$.

## A.3 LINEAR PROGRAMMING

Linear programming is traditionally associated to constrained optimization problems, but it can be used to define a propositional language $\mathcal{F}_{LP} := (\mathcal{T}_{LP}, \jmath_{LP})$ naturally suited to express many real-world problems.

A theory $\Pi \in \mathcal{T}_{LP}$, called a **linear program**, is a set of formulas called **linear constraints**. A linear constraint $r \in \mathcal{LC}(\mathbf{Y})$ is of the shape:

$$b_1.Y_{i_1} + ... + b_m.Y_{i_m} \leqslant c$$

with $Y_{i_1}, ..., Y_{i_m} \in \mathbf{Y}$ and $b_1, ..., b_m, c \in \mathbb{Z}$.

To lighten notations, a linear constraint is sometimes noted $\langle \mathbf{Z}, \mathbf{b} \rangle = c$ where $\mathbf{Z} := (Y_{i_1}, ..., Y_{i_m})$ and $\mathbf{b} := (b_1, ..., b_m)$.

*Remark* 3. The set of coefficients $\mathbb{Z}$ can be extended to $\mathbb{Q}$ without affecting either the concision of the language nor its expressivity. However, it cannot be extended to $\mathbb{R}$ because arbitrary irrational coefficients would require an infinite length to represent.

A state $\mathbf{y} \in \mathbb{B}^{\mathbf{Y}}$ **satisfies** a linear constraint iff $b_1.y_{i_1} + ... + b_m.y_{i_m} \leqslant c$ in the usual arithmetical sense. A state $\mathbf{y} \in \mathbb{B}^{\mathbf{Y}}$ satisfies a linear program $\Pi \in \mathcal{T}_{LP}$ (*i.e.*, $\jmath_{LP}(\Pi)(\mathbf{y}) = 1$) iff it satisfies all linear constraints in $\Pi$.

**Example 6.** *Imagine a catalog of products* $\mathbf{P} := \{P_1, ..., P_k\}$ *with corresponding prices* $\mathbf{p} := \{p_1, ..., p_k\} \in \mathbb{N}^k$. *A basket of products corresponds to a state on* $\mathbf{P}$. *An online website might want to suggest a basket of additional products to go with the order of a client. However, it noticed that large or expansive baskets are less likely to be picked. However, they would like to make sure that the suggested basket is not too cheap. Therefore, they defined a maximum size* $N_M$ *as well as maximum and minimum budgets* $B_M$ *and* $B_m$ *for the suggested baskets. The set of baskets that match those constraints correspond to a boolean function on* $\mathbf{P}$.

*This boolean function can be represented by the following linear program:*

$$\langle \mathbf{P}, \mathbf{1} \rangle \leqslant N_M$$
$$\langle \mathbf{P}, \mathbf{p} \rangle \leqslant B_M$$
$$\langle \mathbf{P}, \mathbf{p} \rangle \geqslant B_m$$

### A.4  GRAPH-BASED LANGUAGES

Although they are not usually thought of as propositional languages, graphs can allow to express knowledge about variables by relating them to elements of a graph. As opposed to most propositional languages, graph-based languages are not universal (*i.e.*, they cannot represent any boolean function), but specialized for a specific type of knowledge. They are often used to represent fragments of universal languages in a concise and more intuitive way. Besides, unlike most propositional languages where the semantic is strongly tied to the syntax (such that the semantics is often implicitly assumed from the syntax), graph-based languages share very similar syntaxes but vary greatly in their semantics.

A language is **edge-based** (resp. **vertice-based**) when a theory maps variables in $\mathbf{Y}$ to edges (resp. vertices) of a graph $G = (V, E)$: a theory is a couple $T := (G, \varsigma)$ where $G = (V, E)$ is a graph and $\varsigma : E \to \mathbf{Y}$ (resp. $\varsigma : V \to \mathbf{Y}$) is **bijective**.

**Example 7.** *The simple path language (resp. matching language) is an edge-based language where a state* $\mathbf{y} \in \mathbb{B}^{\mathbf{Y}}$ *satisfies* $T$ *iff the set of selected edges forms a simple path (resp. perfect matching) in* $G$.

## B  PROPERTIES

We give formal definitions and proofs for syntactic invariance and consistency of neurosymbolic techniques mentioned in the paper. Besides, we demonstrate several other properties of interest of probabilistic neurosymbolic techniques. We underline the generality of semantic conditioning by showing that traditional loss and inference modules introduced for independent, categorical and hierarchical classification are specific cases of semantic conditioning on their respective semantics. We demonstrate that conditioning at inference is *superior* to independent inference in a certain sense that we define.

### B.1 SYNTACTIC INVARIANCE

**Definition 9** (Syntactic invariance). A model agnostic neurosymbolic technique $\mathfrak{T} := (\mathfrak{L}, \mathfrak{I})$ for a propositional language $\mathcal{F} := (\mathcal{T}, \jmath)$ is **invariant to syntax** iff, for any finite set of variables $\mathbf{Y}$ and theories $\kappa_1, \kappa_2 \in \mathcal{T}(\mathbf{Y})$ such that $\kappa_1 \equiv \kappa_2$:

$$\mathfrak{L}(\mathbf{Y}, \kappa_1) = \mathfrak{L}(\mathbf{Y}, \kappa_2)$$
$$\mathfrak{I}(\mathbf{Y}, \kappa_1) = \mathfrak{I}(\mathbf{Y}, \kappa_2)$$

**Proposition 1.** *Semantic regularization, semantic conditioning and semantic conditioning at inference are all invariant to syntax.*

*Proof.* Like mentioned in the paper, this stems directly from the fact that probabilistic reasoning essentially depends on the semantic of the formula rather than on its syntax. By definition, for $\kappa_1 \equiv \kappa_2$ we have $\mathbb{1}_{\kappa_1} = \mathbb{1}_{\kappa_2}$. This implies that for any $\mathbf{a} \in \mathbb{R}^k$ we have $\mathcal{P}(\kappa_1|\mathbf{a}) = \mathcal{P}(\kappa_2|\mathbf{a})$ and $\arg\max_{\mathbf{y} \in \mathbb{B}^{\mathbf{Y}}} \mathcal{P}(\cdot|\mathbf{a}, \kappa_1) = \arg\max_{\mathbf{y} \in \mathbb{B}^{\mathbf{Y}}} \mathcal{P}(\cdot|\mathbf{a}, \kappa_2)$, which concludes the proof for all three techniques. □

### B.2 CONSISTENCY

**Definition 10** (Consistency). A model agnostic neurosymbolic technique $\mathfrak{T} := (\mathfrak{L}, \mathfrak{I})$ for a propositional language $\mathcal{F} := (\mathcal{T}, \jmath)$ is **consistent** iff, for any finite set of variables $\mathbf{Y}$ and a satisfiable theory $\kappa \in \mathcal{T}(\mathbf{Y})$:

$$\forall \mathbf{a} \in \mathbb{R}^k, \mathfrak{I}(\mathbf{Y}, \kappa)(\mathbf{a}) \models \kappa$$

**Proposition 2.** *Both semantic conditioning and semantic conditioning at inference are consistent.*

*Proof 2.* Remind that $\mathfrak{I}_{sc}(\mathbf{Y}, \kappa)(\mathbf{a}) = \arg\max_{\mathbf{y} \in \mathbb{B}^{\mathbf{Y}}} \mathcal{P}(\mathbf{y}|\mathbf{a}, \kappa)$.

We assumed $\kappa$ to be satisfiable and we know that for all $\mathbf{a}$, $\mathcal{P}(\cdot|\mathbf{a})$ is strictly positive. Therefore $\mathcal{P}(\cdot|\mathbf{a}, \kappa)$ is strictly positive and we have:

$$\mathbf{y} = \arg\max_{\mathbf{y} \in \mathbb{B}^{\mathbf{Y}}} \mathcal{P}(\mathbf{y}|\mathbf{a}, \kappa) \implies \mathcal{P}(\mathbf{y}|\mathbf{a}, \kappa) > 0 \implies \mathbf{y} \models \kappa$$

Hence:

$$\forall \mathbf{a}, \mathfrak{I}_{sc}(\mathbf{Y}, \kappa)(\mathbf{a}) \models \kappa$$

□

### B.3 GENERALITY OF SEMANTIC CONDITIONING

First, let us demonstrate that standard modules for independent and categorical classification are particular cases of semantic conditioning on their respective background knowledge:

**Proposition 3.**

$$\begin{align} \mathfrak{L}_{sc}(\mathbf{Y}, \top)(\mathbf{a}, \mathbf{y}) = \mathsf{L}_{imc}(\mathbf{a}, \mathbf{y}) \quad & \mathfrak{I}_{sc}(\mathbf{Y}, \top)(\mathbf{a}) = \mathsf{I}_{imc}(\mathbf{a}) \\ \mathfrak{L}_{sc}(\mathbf{Y}, \kappa_{\bigodot_k})(\mathbf{a}, \mathbf{y}) = \mathsf{L}_{\bigodot_k}(\mathbf{a}, \mathbf{y}) \quad & \mathfrak{I}_{sc}(\mathbf{Y}, \kappa_{\bigodot_k})(\mathbf{a}) = \mathsf{I}_{\bigodot_k}(\mathbf{a}) \end{align} \tag{16}$$

We start by demonstrating the following lemma:

**Lemma 4.** *Let's assume $\mathbf{a} \in \mathbb{R}^k$, then:*

$$\mathcal{P}(\mathbf{y}|\mathbf{a}) = \prod_{1 \leqslant j \leqslant k} y_j.\mathsf{s}(a_j) + (1 - y_j).(1 - \mathsf{s}(a_j))$$

*where $\mathsf{s}(\mathbf{a}) = (\frac{e^{a_j}}{1+e^{a_j}})_{1 \leqslant j \leqslant k}$ is the sigmoid function.*

*Proof.* To prove this, let's prove by recurrence on $k \in \mathbb{N}^*$ that:

$$\forall \mathbf{a} \in \mathbb{R}^k, \mathsf{Z}(\mathcal{E}(\cdot|\mathbf{a})) = \prod_{1 \leqslant j \leqslant k} (1 + e^{a_j})$$

First, let's assume $k = 1$, we have:

$$\forall a \in \mathbb{R}, \mathsf{Z}(\mathcal{E}(\cdot|a)) = \mathcal{E}(0|a) + \mathcal{E}(1|a) = e^0 + e^a = 1 + e^a$$

Then, let's assume $k > 1$, we have:

$$\forall \mathbf{a} \in \mathbb{R}^k, \mathsf{Z}(\mathcal{E}(\cdot|\mathbf{a})) = \sum_{\mathbf{y} \in \mathbb{B}^{\mathbf{Y}}} \mathcal{E}(\mathbf{y}|\mathbf{a}) = \sum_{\mathbf{y} \in \mathbb{B}^{\mathbf{Y}}} \prod_{1 \leqslant i \leqslant k} e^{a_i \cdot y_i}$$

$$= \sum_{\substack{\mathbf{y} \in \mathbb{B}^{\mathbf{Y}} \\ y_k = 0}} \prod_{1 \leqslant i \leqslant k-1} e^{a_i \cdot y_i} + \sum_{\substack{\mathbf{y} \in \mathbb{B}^{\mathbf{Y}} \\ y_k = 1}} e^{a_k} \cdot \prod_{1 \leqslant i \leqslant k-1} e^{a_i \cdot y_i}$$

$$= (1 + e^{a_k}) \cdot \sum_{\mathbf{y} \in \mathbb{B}^{k-1}} \prod_{1 \leqslant i \leqslant k-1} e^{a_i \cdot y_i} = (1 + e^{a_k}) \cdot \mathsf{Z}(\mathcal{E}(\cdot|\mathbf{a}_{\backslash k}))$$

where $\mathbf{a}_{\backslash k} = (a_j)_{1 \leqslant j \leqslant k-1}$.

By application of the recurrence hypothesis:

$$\mathsf{Z}(\mathcal{E}(\cdot|\mathbf{a}_{\backslash k})) = \prod_{1 \leqslant j \leqslant k-1} (1 + e^{a_j})$$

Hence:

$$\forall \mathbf{a} \in \mathbb{R}^k, \mathsf{Z}(\mathcal{E}(\cdot|\mathbf{a})) = \prod_{1 \leqslant j \leqslant k} (1 + e^{a_j})$$

This gives us:

$$\forall \mathbf{y} \in \mathbb{B}^{\mathbf{Y}}, \forall \mathbf{a} \in \mathbb{R}^k, \mathcal{P}(\mathbf{y}|\mathbf{a}) = \frac{\mathcal{E}(\mathbf{y}|\mathbf{a})}{\mathsf{Z}(\mathcal{E}(\cdot|\mathbf{a}))} = \frac{\prod_{1 \leqslant i \leqslant k} e^{a_i \cdot y_i}}{\prod_{1 \leqslant j \leqslant k}(1 + e^{a_j})} = \prod_{1 \leqslant j \leqslant k} \frac{e^{a_i \cdot y_i}}{1 + e^{a_j}}$$

Notice that:

$$\forall y \in \mathbb{B}, a \in \mathbb{R}, \frac{e^{a \cdot y}}{1 + e^a} = y.\mathsf{s}(a) + (1 - y).(1 - \mathsf{s}(a))$$

Thus, finally:

$$\forall \mathbf{y} \in \mathbb{B}^{\mathbf{Y}}, \forall \mathbf{a} \in \mathbb{R}^k, \mathcal{P}(\mathbf{y}|\mathbf{a}) = \prod_{1 \leqslant j \leqslant k} y_j.\mathsf{s}(a_j) + (1 - y_j).(1 - \mathsf{s}(a_j))$$

$\square$

*Proof 3.1.* First, according to Lemma 4:

$$\mathcal{P}(\mathbf{y}|\mathbf{a}) = \prod_{1 \leqslant j \leqslant k} y_j.\mathsf{s}(a_j) + (1 - y_j).(1 - \mathsf{s}(a_j))$$

Besides, we know that $\mathbb{1}_\top = 1$ (all states are mapped to 1), which implies that:

$$\forall \mathbf{y} \in \mathbb{B}^{\mathbf{Y}}, \forall \mathbf{a} \in \mathbb{R}^k, \mathcal{P}(\mathbf{y}|\mathbf{a}, \top) = \mathcal{P}(\mathbf{y}|\mathbf{a})$$

This gives:

$$\mathfrak{L}_{sc}(\mathbf{Y}, \top)(\mathbf{a}, \mathbf{y}) = -\log(\mathcal{P}(\mathbf{y}|\mathbf{a}, \top)) = -\log(\mathcal{P}(\mathbf{y}|\mathbf{a}))$$

$$= -\log(\prod_j y_j.\mathsf{s}(a_j) + (1 - y_j).(1 - \mathsf{s}(a_j)))$$

$$= -\sum_j \log(y_j.\mathsf{s}(a_j) + (1 - y_j).(1 - \mathsf{s}(a_j)))$$

Since $\mathbf{y}$ is a binary vector:

$$\mathfrak{L}_{sc}(\mathbf{Y}, \top)(\mathbf{a}, \mathbf{y}) = -\sum_j y_j.\log(\mathsf{s}(a_j)) + (1 - y_j).\log(1 - \mathsf{s}(a_j))$$

$$= \mathsf{L}_{imc}(\mathbf{a}, \mathbf{y})$$

$\square$

*Proof 3.2.*

$$\mathfrak{I}_{sc}(\mathbf{Y}, \top)(\mathbf{a}) = \underset{\mathbf{y} \in \mathbb{B}^{\mathbf{Y}}}{\arg\max} \mathcal{P}(\mathbf{y}|\mathbf{a}, \top) = \underset{\mathbf{y} \in \mathbb{B}^{\mathbf{Y}}}{\arg\max} \mathcal{P}(\mathbf{y}|\mathbf{a}) = \underset{\mathbf{y} \in \mathbb{B}^{\mathbf{Y}}}{\arg\max} \mathcal{E}(\mathbf{y}|\mathbf{a})$$

$$= \underset{\mathbf{y} \in \mathbb{B}^{\mathbf{Y}}}{\arg\max} \prod_{1 \leqslant i \leqslant k} e^{a_i \cdot y_i} = \underset{\mathbf{y} \in \mathbb{B}^{\mathbf{Y}}}{\arg\max} [\exp(\sum_{1 \leqslant i \leqslant k} a_i.y_i)] = \underset{\mathbf{y} \in \mathbb{B}^{\mathbf{Y}}}{\arg\max} \sum_{1 \leqslant i \leqslant k} a_i.y_i$$

$$= \mathbf{1}[\mathbf{a} \geqslant 0]$$

$$= \mathsf{I}_{imc}(\mathbf{a})$$

$\square$

*Proof 3.3.* The one and only one semantic of $\kappa_{\odot_k}$ gives us:

$$\forall \mathbf{y}, \mathbf{y} \models \kappa_{\odot_k} \implies \exists j, \mathbf{y} = \odot_k(j)$$

Hence:

$$\mathcal{P}(\kappa_{\odot_k}|\mathbf{a}) = \sum_{\mathbf{y} \models \kappa_{\odot_k}} \mathcal{P}(\mathbf{y}|\mathbf{a}) = \sum_{1 \leqslant j \leqslant k} \mathcal{P}(\odot_k(j)|\mathbf{a})$$

$$= \frac{1}{\mathsf{Z}(\mathcal{P}(\cdot|\mathbf{a}))} \cdot \sum_{1 \leqslant j \leqslant k} \mathcal{E}(\odot_k(j)|\mathbf{a}) = \frac{1}{\mathsf{Z}(\mathcal{P}(\cdot|\mathbf{a}))} \cdot \sum_{1 \leqslant j \leqslant k} e^{a_j}$$

This leads to:

$$\forall l, \mathcal{P}(\odot_k(l)|\mathbf{a}, \kappa_{\odot_k}) = \frac{\mathcal{P}(\odot_k(l) \wedge \kappa_{\odot_k}|\mathbf{a})}{\mathcal{P}(\kappa_{\odot_k}|\mathbf{a})} = \frac{\mathcal{P}(\odot_k(l)|\mathbf{a})}{\mathcal{P}(\kappa_{\odot_k}|\mathbf{a})} = \frac{\mathcal{E}(\odot_k(l)|\mathbf{a})}{\mathsf{Z}(\mathcal{P}(\cdot|\mathbf{a})).\mathcal{P}(\kappa_{\odot_k}|\mathbf{a})}$$

$$= \frac{e^{a_l}}{\sum_{1 \leqslant j \leqslant k} e^{a_j}} = \sigma(\mathbf{a})_l = \langle \sigma(\mathbf{a}), \odot_k(l) \rangle$$

Besides, since we assume consistent labels, we know that there is $l$ such that $\mathbf{y} = \odot_k(l)$, which gives:

$$\mathfrak{L}_{sc}(\mathbf{Y}, \kappa_{\odot_k})(\mathbf{a}, \mathbf{y}) = \mathsf{L}_{\kappa_{\odot_k}}(\mathbf{a}, \odot_k(l)) = -\log(\mathcal{P}(\odot_k(l)|\mathbf{a}, \kappa_{\odot_k}))$$

$$= -\log(\langle \sigma(\mathbf{a}), \odot_k(l) \rangle) = -\log(\langle \sigma(\mathbf{a}), \mathbf{y} \rangle)$$

$$= \mathsf{L}_{\odot_k}(\mathbf{a}, \mathbf{y})$$

$\square$

*Proof 3.4.* We know that $\kappa_{\odot_k}$ is satisfiable and $\mathcal{P}(\mathbf{y}|\mathbf{a})$ is strictly positive. So we have:

$$\mathbf{y} = \underset{\mathbf{y} \in \mathbb{B}^{\mathbf{Y}}}{\arg\max} \mathcal{P}(\mathbf{y}|\mathbf{a}, \kappa_{\odot_k}) \implies \mathbf{y} \models \kappa_{\odot_k}$$

$$\implies \exists l, \mathbf{y} = \odot_k(l)$$

Therefore, we have:

$$\mathfrak{I}_{sc}(\mathbf{Y}, \kappa_{\odot_k})(\mathbf{a}) = \underset{1 \leqslant j \leqslant k}{\arg\max} \mathcal{P}(\odot_k(j)|\mathbf{a}, \kappa_{\odot_k}) = \underset{1 \leqslant l \leqslant k}{\arg\max} \mathcal{P}(\odot_k(l)|\mathbf{a})$$

$$= \underset{1 \leqslant l \leqslant k}{\arg\max} \langle \mathbf{a}, \odot_k(l) \rangle = \odot_k(\underset{1 \leqslant l \leqslant k}{\arg\max}(\mathbf{a}))$$

$$= \mathsf{I}_{\odot_k}(\mathbf{a})$$

$\square$

## B.4 SUPERIORITY OF CONDITIONING AT INFERENCE

Besides, when performing inference based on identical model modules and learned parameters, *sci* **guarantees** greater or equal accuracy compared to traditional *imc* inference (*i.e.*, if $\mathsf{I}_{imc}$ infers the right labels, then $\mathfrak{I}_{sc}(\mathbf{Y}, \kappa)$ will also infer the right labels):

**Proposition 5.**
$$\forall \mathbf{a} \in \mathbb{R}^k, \mathbf{y} \models \kappa, \mathsf{I}_{imc}(\mathbf{a}) = \mathbf{y} \implies \mathfrak{I}_{sci}(\mathbf{Y}, \kappa)(\mathbf{a}) = \mathbf{y}$$

*Proof 5.* Let's proove this by the absurd and assume that:
$$\mathfrak{I}_{sc}(\mathbf{Y}, \kappa)(\mathbf{a}) := \hat{\mathbf{y}} \neq \mathbf{y}$$

Since both $\hat{\mathbf{y}}$ and $\mathbf{y}$ are consistent with $\kappa$ (which we assume satisfiable), we have:
$$\frac{\mathcal{P}(\hat{\mathbf{y}}|\mathbf{a}, \kappa)}{\mathcal{P}(\mathbf{y}|\mathbf{a}, \kappa)} = \frac{\mathcal{P}(\hat{\mathbf{y}}|\mathbf{a})}{\mathcal{P}(\mathbf{y}|\mathbf{a})}$$

Because $\hat{\mathbf{y}} = \mathfrak{I}_{sc}(\mathbf{Y}, \kappa)(\mathbf{a}) = \underset{\mathbf{y} \in \mathbb{B}^{\mathbf{Y}}}{\arg\max} \mathcal{P}(\mathbf{y}|\mathbf{a}, \kappa)$:
$$\mathcal{P}(\hat{\mathbf{y}}|\mathbf{a}, \kappa) \geqslant \mathcal{P}(\mathbf{y}|\mathbf{a}, \kappa)$$

Therefore:
$$\mathcal{P}(\hat{\mathbf{y}}|\mathbf{a}) \geqslant \mathcal{P}(\mathbf{y}|\mathbf{a}) \implies \mathbf{y} \neq \underset{\mathbf{y} \in \mathbb{B}^{\mathbf{Y}}}{\arg\max} \mathcal{P}(\mathbf{y}|\mathbf{a}) = \mathsf{I}_{imc}(\mathbf{a})$$

Which is in contradiction with our premise, thus we have
$$\forall \mathbf{a} \in \mathbb{R}^k, \mathbf{y} \models \kappa, \mathsf{I}_{imc}(\mathbf{a}) = \mathbf{y} \implies \mathfrak{I}_{sci}(\mathbf{Y}, \kappa)(\mathbf{a}) = \mathbf{y}$$

$\square$

### B.5 RELATION BETWEEN SEMANTIC REGULARIZATION AND CONDITIONING

Finally, it is interesting to notice that under the consistent label hypothesis:

**Proposition 6.**
$$\mathfrak{L}_{sc}(\mathbf{Y}, \kappa)(\mathbf{a}, \mathbf{y}) = -\log(\mathcal{P}(\mathbf{y}|\mathbf{a})) + \log(\mathcal{P}(\kappa|\mathbf{a})) = \mathfrak{L}_{sr}^{-1}(\mathbf{Y}, \kappa)(\mathbf{a}, \mathbf{y}) \tag{17}$$

*Proof 6.* Since labels are consistent, we have $\mathbb{1}_\kappa(\mathbb{1}_\kappa) = 1$, thus $\mathcal{P}(\mathbf{y}|\mathbf{a}, \kappa) = \frac{\mathcal{P}(\mathbf{y}|\mathbf{a})}{\mathcal{P}(\kappa|\mathbf{a})}$.

Therefore:
$$\begin{aligned}
\mathfrak{L}_{sc}(\mathbf{Y}, \kappa)(\mathbf{a}, \mathbf{y}) &= -\log(\mathcal{P}(\mathbf{y}|\mathbf{a}, \kappa)) = -\log(\frac{\mathcal{P}(\mathbf{y}|\mathbf{a})}{\mathcal{P}(\kappa|\mathbf{a})}) \\
&= -\log(\mathcal{P}(\mathbf{y}|\mathbf{a})) + \log(\mathcal{P}(\kappa|\mathbf{a})) \\
&= \mathfrak{L}_{sr}^{-1}(\mathbf{Y}, \kappa)(\mathbf{a}, \mathbf{y})
\end{aligned}$$

$\square$

Thus, the loss module of semantic conditioning corresponds to that of semantic regularization with a $\lambda = -1$. Although it seems counter-intuitive that two systems trying to reach the same goal end up using "opposite regularization terms" in their loss module, this is justified by the different inference modules used in each system.

Hence, an implementation for $\mathfrak{L}_{sr}^\lambda(\mathbf{Y}, \kappa)$ can be used for $\mathfrak{L}_{sc}(\mathbf{Y}, \kappa)$. Besides, by training systems with regularized loss modules with different $\lambda$ and evaluating with both $\mathfrak{I}_{sc}(\mathbf{Y}, \kappa)$ and $\mathsf{I}_{imc}$, we can span the entire spectrum of techniques in *imc*, *sr*, *sc* and *sci*.

## C EXPERIMENTAL DETAILS

### C.1 TASKS

We give additional details on the tasks tackled in the paper.

### C.1.1 CATEGORICAL CLASSIFICATION

We mentioned earlier (see Section 3) how categorical classification tasks could be framed as a multi-label classification with prior knowledge. MNIST is one of the oldest and most popular dataset in computer vision and consists of small images of hand-written digits (*e.g.* ⬛ or ⬛). Since its introduction in LeCun et al. (1998), it has been used as a *toy* dataset in many different settings. Likewise in neurosymbolic literature, many researchers used MNIST as a basis to build structured dataset compositionally (*e.g.* the PAIRS dataset in Marra et al. (2020), the MNIST-Add dataset in Manhaeve et al. (2021); Badreddine et al. (2022); Maene & De Raedt; van Krieken et al. or the Sudoku dataset in Augustine et al. (2022)).

### C.1.2 HIERARCHICAL CLASSIFICATION

The Cifar-100 dataset (Krizhevsky, 2009) is composed of 60,000 images classified into 20 mutually exclusive super-classes (*e.g. reptile*), each divided into 5 mutually exclusive fine-grained classes (*e.g. crocodile*, *dinosaur*, *lizard*, *turtle*, and *snake*). While most papers only consider the categorical classification task arising form the 100 fine-grained classes, we keep all 120 classes to produce a multi-label classification task where prior knowledge captures mutual exclusion and the hierarchy between super and fine-grained classes.

The ImageNet Large Scale Visual Recognition Challenge (ILSVRC) (Russakovsky et al., 2015) is an image classification challenge which has become a standard benchmark in computer vision to compare performances of deep learning models. As of August 2014, ImageNet contained 14,197,122 annotated images organized into 21,841 synsets of the WordNet hierarchy (Miller, 1995), however standard image classification tasks often use a subset of those, usually 1,000 or 100 synsets. The WordNet hierarchy defines subsumption (or inclusion) between classes, and can be used in many ways to create a task of binary multi-label classification with prior knowledge.

For our experiments, we sample 100 classes from 1k ImageNet and add all their parent classes. We then prune classes that have only one parent class and one child class to avoid classes having identical sample sets. We thus obtain a dataset of ImageNet samples labeled on a hierarchy of 271 classes. Prior knowledge for this task includes the hierarchical knowledge coming from WordNet, as well as exclusion knowledge that we obtain by assuming two classes having no common descendants are mutually exclusive.

### C.1.3 SIMPLE PATH PREDICTION

The Warcraft shortest path task (Pogančić et al., 2019; Yang et al.; Niepert et al., 2021; Ahmed et al., 2022a) uses randomly generated images of terrain maps from the Warcraft II tileset. Maps are build on a $12 \times 12$ directed grid (each vertex is connected to all its *neighbors*) and to each vertex of the grid corresponds a tile of the tileset. Each tile is a RGB image that depicts a specific terrain, which has a fixed traveling cost. For each map, the label encodes the shortest s-t path (*i.e.*, a path from the upper-left to the lower-right corners), where the weight of the path is the sum of the traveling costs of all terrains (*i.e.*, grid vertices) on the path. The terrain costs are used to produce the dataset but are not provided during training nor inference. In the original dataset (Pogančić et al., 2019), output variables correspond to vertices in the grid and a state satisfies the simple path constraint if the vertices set to 1 constitute a simple s-t path.

This representation comes with several issues. First, as noted in Ahmed et al. (2022a), the set of vertices ambiguously encode more than one path (because of cycles in the grid, there are several possible simple paths that go through the same vertices). Besides, computing MPE and PQE for simple path constraints on general directed graphs are respectively NP-hard (Karp) and #P-hard (Valiant). To make this task tractable, (Ahmed et al., 2022a) transforms the output space in the following way: edges of the grid are chosen as output variables instead of vertices and only simple paths with a maximal length of 29 (the maximal length found in the training set) are kept. This implies that the test set might not be consistent with the constraints since it might contain a path longer than 29 edges. Besides, such method would not scale to larger grids.

In our experiments, we adopt a different approach. We keep the set of edges as our output variables, but we turn the grid into an acyclic graph by only connecting vertices to their right and lower neighbors. Acyclicity is a sufficient condition to compile simple path constraints to an Ordered Bi-

nary Decision Diagram (see `code/circuits/AcyclicSimplePath.py` in the supplementary material), which makes MPE and PQE tractable. This transformation allows us to scale to larger grids without an explosion of the computational cost. We recompute the labels for the new output space using the terrain costs.

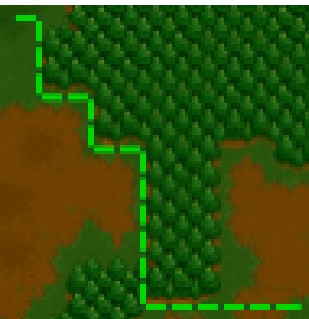 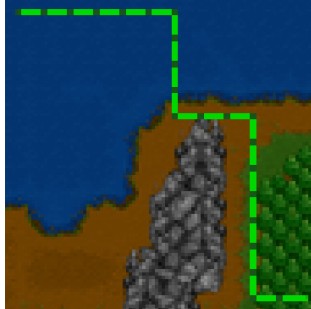

Figure 4: Examples of Warcraft maps and their shortest paths (Ahmed et al., 2022a)

## C.2 IMPLEMENTATION

We used a simple Convolutional Neural Network (CNN) (LeCun et al., 1998) design on the MNIST dataset and the family of DenseNets (Huang et al., 2017) on all others.

In all our experiments, probabilistic reasoning computations brought no significant overhead on top of the neural network. For categorical tasks, the number of valid states is enumerable in linear time, hence there are no complexity issues to implement all three techniques. For hierarchical tasks, we implemented our own version [1] of (Deng et al., 2014), which uses a custom compilation algorithm to convert the propositional formula into a minimal junction tree and then applies a sparse message passing procedure. For simple path prediction task, we used the compilation technique mentioned above alongside the SPL package (Ahmed et al., 2022a) to implement semantic conditioning and semantic regularization. For semantic conditioning at inference, we simply adapted a shortest path solver from NetworkX (Hagberg et al., 2008) based on the Bellman-Ford algorithm (Bellman).

### C.2.1 CATEGORICAL CLASSIFICATION

The architectural design for categorical classification on MNIST is a simple Convolutional Neural Network (CNN) (LeCun et al., 1998), as shown on Listing 1. We trained networks with `num_layers` from 1 up to 9 layers.

Listing 1: Our TinyNet architecture (PyTorch implementation)

```python
class TinyCNNs(nn.Module):
    def __init__(self, num_classes: int = 10,
                 num_layers: int = 1,
                 in_channels=1):
        super().__init__()
        convs = []
        for i in range(num_layers):
            convs.append(nn.Conv2d(2**(i//2)*in_channels,
                                   2**((i+1)//2)*in_channels,
                                   5,
                                   padding=2))
            convs.append(nn.ReLU())
        self.convs = nn.Sequential(*convs)
        # self.AdaptativeScale = int(5*2**(num_layers/2))
        self.pool = nn.AdaptiveAvgPool2d(5)
```

---

[1]Their code was not publicly available. Our code is attached in the supplementary materials and will be made publicly available in the final version.

```
1188            self.fc = nn.Linear(25*2**(num_layers//2)*in_channels,
1189                                num_classes)
1190
1191     def forward(self, x):
1192         x = self.convs(x)
1193         x = F.relu(x)
1194         x = self.pool(x)
1195         x = torch.flatten(x, start_dim=1)
1196         x = self.fc(x)
1197         return x
```

Then, to complete the neural based system: we use the loss module shown on Listing 2 (with varying $\lambda$) and inference modules shown in Listings 2 and 3 for *imc* and sci respectively.

Listing 2: *rsc* loss

```
scores = self.model(x)
energies = torch.gather(self.scores, 1, y.unsqueeze(dim=1))
log_z = torch.sum(torch.log(torch.exp(self.scores).add(1)),
                             dim=1, keepdim=True)
mc_log_z = torch.log(torch.sum(torch.exp(self.scores),
                                 dim=1, keepdim=True))

loss = torch.mean(torch.sub(torch.add(log_z.mul(1+self.lambda),
                                      mc_log_z.mul(-self.lambda)),
                            energies))

return loss
```

Listing 3: *imc* inference

```
scores = self.model(x)
return torch.gt(scores, 0)
```

Listing 4: *sci* inference

```
scores = self.model(x)
_, idx_max = torch.max(self.scores, dim=1)
return F.one_hot(idx_max, num_classes=scores.shape[1])
```

### C.2.2 HIERARCHICAL CLASSIFICATION

The architectural design for hierarchical classification tasks Cifar and ImageNet was based on DenseNets (Huang et al., 2017). We used the `torchvision` implementation with a naive scaling strategy to create DenseNets of various `size`, as shown on Listing 5. We trained network with `size` from 0 up to 8.

Listing 5: DenseNet scaling

```
from torchvision.models.densenet import _densenet

network = _densenet(growth_rate=32,
                    block_config=(6, 12, (size+3)*8,(size+1)*8),
                    num_init_features=64,
                    weights=None,
                    progress=True)
```

For the loss and inference modules, we followed (Deng et al., 2014) and expressed the hierarchical and exclusion relations as a HEX-graph $H$, then compiles this HEX-graph into a HEX-layer that can compute $l_{\kappa_H}$ using a sparse max-product message passing algorithm with Viterbi decoding (see `fastHEXLayer.py`).

To do so, for ImageNet, we first extract hierarchical links from the `wn_hyp.pl` file and arrange them into a directed graph. Then, for our experiments, only 100 leaf nodes are randomly sampled from the total 1,000 and the directed graph is trimmed of any node not connected to a sampled leaf node. We also prune nodes that only have one parent and one child to avoid the case where two nodes have the same set of labeled samples (which would make them indistinguishable for the network). This directed graph is fed into a `HEXGraph` object, which computes the sparse and dense version of the hierarchical and exclusion matrices, builds the corresponding junction tree (using the `JunctionTree` object) and records the valid states of each clique and the sum-product matrix of the junction tree. The results of these compilations steps can be saved and loaded: the specific files used in this experiment are `./ImageNet/compilations/100p*`. During training, this `HEXGraph` is loaded from compilation files and passed on to the `HEXLayer` which contains the methods to perform *sci*. The code to perform those compilation steps is found in `ImageNetProcessing.ipynb`.

For Cifar-100, the hierarchy has only two levels (macro and fine-grained classes) and can be retrieved directly online and fed to the `HEXGraph` object.

### C.2.3 SIMPLE PATH PREDICTION

The architectural design for simple path prediction on the Warcraft Shortest Path dataset is also based on DenseNets (Huang et al., 2017) as for hierarchical tasks. We trained network with `size` from 0 up to 6.

We compiled the constraints to an OBDD (Darwiche, 2009) using a custom algorithm. Compilation files can be found in the code under name files `12x12.sdd` and `12x12.vtree` in the `Code/WSP/data/12x12` folder.

Then, we used the SPL (Ahmed et al., 2022a) implementation to compute the loss module for semantic conditioning ($\lambda = -1$) and semantic regularization ($= 0.1$). For the inference module, we decided to replace the SPL implementation by an adapted shortest path solver from NetworkX (Hagberg et al., 2008) based on the Bellman-Ford algorithm (Bellman). We found this solution less prone to numerical stability issues.

### C.3 METRICS

There are plenty of metrics that can be used to evaluate a classification system.

Simple accuracy averages how many classes were correctly labeled on each sample, however, since multi-label classification datasets with background knowledge are often highly unbalanced (far more negative classes that positive ones) it is often unfit to the task. Precision, recall and f1-score metrics can help tackle with this issue, but they lose track of the semantics of the task.

Semantic consistency counts how many outputs are consistent with the background knowledge. Since *sci* is provably consistent, this metric is of little interest for us.

Metrics that are not based on the binary outputs but need to access probability scores associated with each classe, like threshold metrics (*e.g.* map@50, map@75, auc) or top-k metrics, are not accessible to our classification system as is.

Eventually, we decide to use **exact accuracy**, which counts how many samples are perfectly labeled: this is the most demanding metric since a single mistake disqualifies the whole output. This metric is also used in Ahmed et al. (2022a) and in Xu et al. (2018) (where it is called coherent accuracy).

### C.4 HYPERPARAMETERS

**Epochs** We trained each system on the training set for up to 100 epochs: 100 for MNIST, Cifar and Warcraft Shortest Path and only 90 for ImageNet due to computational ressources constraints. We evaluate the perfect accuracy on the test set at each epoch.

**Seeds** We set seeds manually with `torch.manual_seed(args.seed)`. We used 6 seeds ($[0, 1, 2, 3, 4, 5]$) for MNIST, 3 seeds ($[0, 1, 2]$) for Cifar and ImageNet and 2 seeds ($[0, 1]$) for Warcraft Shortest Path (due to compute budget limits).

**Batch size**   We use a batch size of 8 for MNIST and Cifar, and increase to 64 for ImageNet and Warcraft Shortest Path to speed up training.

**Regularization coefficient**   We used $\lambda = 0.1$ for semantic regularization (Xu et al., 2018). We did not perform a full hyperparameter search procedure because we considered it was too costly on large datasets like ImageNet, Cifar or Warcraft Shortest Path. We tried several values of $\lambda$ on MNIST and at the beginning of our experiments on Cifar and ImageNet with no noticeable difference.

**Optimizer**   We use Adam with a learning rate of $10^{-4}$ for all tasks.

