# OpenReview forum: "A bird's eye view on informed classification"
_ICLR.cc/2025/Conference — Submitted to ICLR 2025_

### Official Review · Reviewer_4WvM · 2024-10-24

**Soundness:** 4
**Presentation:** 3
**Contribution:** 1
**Rating:** 3
**Confidence:** 3

**Summary:**

The paper makes two related contributions:

* First, the paper proposes a general framework for encoding prior knowledge, within the context of probabilistic neurosymbolic approaches for multi-label classification.
* Second, the paper proposes a variant on existing methods for  probabilistic neurosymbolic multi-label classification. Specifically, while in existing work, semantic conditioning is applied both at training time and at inference time, the authors propose to do this only at inference time. This has the advantage of making the method much more efficient, while experimental results show that the penalty for doing this only at inference time is (at worst) limited.

**Strengths:**

The paper is carefully written and proposes a clean formalisation of probabilistic neurosymbolic multi-label classification. The formulation clearly illustrates the differences between two important previous techniques: semantic regularization and semantic conditioning.

The fact that performing semantic conditioning only at inference time does not deteriorate the performance much is a useful finding, which is worth highlighting to practitioners.

**Weaknesses:**

The main weakness, in my view, is that the paper only makes a small and incremental contribution. Claiming that "semantic conditioning at inference" is a new technique is somewhat of a stretch. It is essentially an ablation of the full semantic conditioning method.

The other claimed contribution is that the paper provides a unified formalism. To claim this as a contribution is questionable. The paper could easily have used propositional logic as the underlying framework and then have made the point that other propositional languages can be treated similarly. The current formulation also has the drawback of making the notation heavy and the explanations more opaque.

Some minor points:

The last paragraph before section 4.1 (on fuzzy logics) feels out of place, and similar for the first paragraph of Section 4.1. The whole built-up of the framework has assumed probabilistic reasoning, so it seems weird to suddenly justify this choice in the middle of the paper.

Typo in the abstract: theoritical

Figure 2: please add explanations for the different acronyms (SCI, IMC, etc) to the caption, even though their meaning can be guessed.

**Questions:**

For the experiments, the only baseline is to use an uninformed model. At least, I would have expected something like fuzzy regularization there as well.

---

> ### Author Response · Authors · 2024-11-14
> **Baselines and fuzzy regularization**
>
> Regarding baselines, we considered that semantic regularization and semantic conditioning constitued strong baselines to evaluate our contribution semantic conditioning at inference.
>
> Besides, since fuzzy-based techniques are developed for propositional formulas and are not syntax independent, the evaluation of fuzzy regularization on our tasks would have depended on how we to chose to translate prior knowledge into a propositional formula. This choice seemed arbitrary to us and would have required to take some time to justify it.
>
> In fact, a specific translation into propositional formula (to be specific, compilation to the "unfolded" version of a smooth dDNNF) makes fuzzy regularization and semantic regularization equivalent. This means that fuzzy regularization is, in this narrow sense, present in our baselines.
>
> We did not expect another encoding into propositional formulas to make fuzzy regularization more accurate and therefore decided not to explicitely include it into our evaluations.

---

> > ### Comment · Reviewer_4WvM · 2024-11-15
> >
> > Thanks for the clarifications.
> >
> > I understand the arguments, but I still think it would have been useful to add fuzzy regularization. The syntax-dependence is a valid concern, but you could address this by analyzing the variation in performance that arises from using different (classically equivalent) variations of the formula.

---

### Official Review · Reviewer_yzJ2 · 2024-10-24

**Soundness:** 3
**Presentation:** 2
**Contribution:** 1
**Rating:** 3
**Confidence:** 3

**Summary:**

### Paper Summary
The authors introduce syntax/formalism(s) that bridge a number of works interested in informed classification.

The authors find that simplifying Semantic Conditioning (SC) to Semantic Conditioning at Inference (SCI) captures 75% of the improved performance gain. This is particularly valuable because it suggests that we can flexibly use existing models without additional training!

### Reviewer's Note
I am not an expert in neurosymbolic systems. I have read some of these papers, but this not an area of my direct expertise, so it is possible I am missing something.

### Review Summary
The authors argue that introducing the additional syntax does is a contribution in-and-of-itself, but I don't (yet) agree. The shared syntax does not seem to illustrate/highlight/show something new. I would expect that the authors of the mentioned previous works would agree that a shared syntax could be devised across problems, but instead chose to use the most apt approach for their respective formulations. I do think that it may be useful, if it leads to new developments/understandings, but the formalism itself only seems like 1/2 of a contribution.

(I am quite open to discussion on this point or alternative perspectives.)

This leaves the primary contribution of this paper being: Removing part of an existing method only partially reduces performance. This is good to know, but I'm not sure its critical. Another way of looking at this is that I think the paper spends a lot of energy/time presenting this new syntax but I don't think we (the reader/the field) is getting a lot out of this, and the contribution of SC --> SCI is not particularly/directly related to the syntax.

**Strengths:**

* The paper does a great job of providing a lot of background and previous work.
    * So much so that this paper feels mostly like an expositionary piece. If ICLR is interested in exposition-only work (somewhat like a focused survey) and the paper were slightly modified to pull this forward, then I would find the work more compelling. (I don't think that this is the case?)

**Weaknesses:**

* The paper spends a lot of time basically defining an (existing) task and a syntax for this task; I didn't find this to be particularly compelling.
* I think a paper that really leaned into this SC --> SCI contribution and included qualitative/quantitative analysis of what/how these models differ and what mistakes are made by the respective models that that could be a strong paper. As it stands, I am currently unconvinced.

### Nitpicks
* I found the introduction on what a deep learning system in section 3.1 a bit strange / out of place.
* line 128: "is build by" --> "is built by", or else, reword the sentence
* figure 1: labels part of the system as a NeSy but this is never mentioned anywhere else. (NeuroSymbolic Technique). I think there is room for the full definition in the figure, or else, include the abbreviation in the text.
* figure 2 is a bit hard to read/small. try to resize this figure? use different settings or software to create it?
* line 483: "elucidate" --> "adjudicate"

**Questions:**

What did the new syntax exactly offer/contribute to your understanding/method/results? Am I missing or misunderstanding something?

While the background is already taking up a lot of the paper, finding space to provide key examples of the type of data you're working with could help get the reader to quickly track what you want to investigate.

Relatedly, what errors does SCI make that SC doesn't? Can you provide examples of this?

---

### Official Review · Reviewer_mFsY · 2024-10-31

**Soundness:** 3
**Presentation:** 3
**Contribution:** 2
**Rating:** 3
**Confidence:** 2

**Summary:**

The paper takes a neurosymbolic AI approach to multi-label classification where constraints of the output space are imposed using a propositional language and these constraints are accounted for when making predictions. It provides extensive background material on the formalisms that the method employs and evaluates on some multi-label classification benchmarks, demonstrating that accounting for these constraints outperforms not accounting for them.

**Strengths:**

I appreciate that your paper employs techniques that are not particularly popular in mainstream ML. It's great to bring exposure to alternative ideas.

**Weaknesses:**

One challenge of employing techniques that are less familiar to readers is that the paper needs to set up a significant amount of background material. This occupies the majority of the paper. Some of the content is fairly elementary, such as explaining how to use a differentiable loss function to learn a probabilistic classifier.

The experiments are not particularly rigorous. I would have appreciated more ablations analyzing, for example, the impact of different algorithmic choices. For example, what fraction of predictions actually violate the constraints for the baseline model? There is also no benchmarking against other approaches to multi-label classification from prior works that go beyond independent prediction across labels.

**Questions:**

It's important to bridge the gap between the neurosymbolic AI community and related ideas that have previously appeared in other parts of the literature. A significant missing piece in the discussion of related work is anything regarding probabilistic graphical models, which are based on similar formalisms as section 2.

Based on some quick background reading, here are some key multi-label classification papers that did inference in undirected graphical models:

Ghamrawi, Nadia and McCallum, Andrew. Collective multi-label classification.
Finley, Thomas and Joachims, Thorsten. Training structural svms when exact inference is intractable.
Meshi, Ofer, Sontag, David, Globerson, Amir, and Jaakkola, Tommi S. Learning efficiently with approximate inference via dual losses
Petterson, James and Caetano, Tiberio S. Submodular multi-label ´ learning.

Can you please discuss more directly the relationship to PGMs? For the particular problem setups you use, is there a corresponding formulation as a PGM?

=========
I had a hard time understanding the complexity of the actual prediction problems. It would have been great if you had compressed some of the background and instead provide concrete details about the structure of the constraints in each problem that is considered in the experiments and how inference wrt these constraints is handled. I know some of these are provided in the appendix. Can you provide a high-level summary here, such that such a summary could be included in the main paper?

---

> ### Author Response · Authors · 2024-11-14
> **Relation to PGMs**
>
> There are interesting ties between our work and PGMs.
>
> First, the conditioned distribution on which semantic conditioning is based can be captured by a PGM when the prior knowledge is expressed as a CNF formula: the primal graph of the CNF (ie. the graph that has a node for each variable and an edge between two nodes if both variables appear in the same clause) defines an undirected graphical model that captures the conditioned distribution. In such case, the loss and inference modules of semantic conditioning simply correspond to maximum likelihood estimation and mode estimation of the PGM, and standard sum-product and max-sum message passing algorithms on PGMs can be used to perform PQE and MPE. In fact, the work on Large-Scale Object Classification using Label Relation Graphs by Jia Deng et al. mentioned in our paper can be seen as a semantic conditioning technique based on PGMs and custom message passing algorithms.
>
> Moreover, when prior knowledge is expressed in CNF, message passing algorithms can also be used to compute PQE for semantic regularization and MPE for semantic conditioning at inference even though the probabilistic interpretation of the system is no longer valid.
>
> However, there are limits to these parrallels.
>
> First, when prior knowledge is not expressed in CNF, or another language based on conjunctions of constraints, there is no straightforward way to design a PGM that captures the conditioned distribution. In this sense, our formalism differs from PGM since the distribution is defined through prior knowledge and not through interaction or independence relations like in PGMs.
>
> Finally, standard message passing algorithms mainly exploit tree decompositions to perform PQE and MPE and are therefore tractable for bounded tree-width formulas. This bounded tree-width condition is quite restrictive. Knowledge compilation methods, which go beyond this criteria and enable to perform PQE and MPE for a larger class of formulas, have therefore become the dominant approach in the neurosymbolic literature.

---

> > ### Comment · Reviewer_mFsY · 2024-11-19
> >
> > Thanks for the details. Can you please explain the particular set of formulas used for each of the problems in your experiments? Are these in the broader set of formulas that a PGM couldn't represent? I'm trying to understand if the concrete use cases in your paper are qualitatively different from a PGM, not if the general framework you present is qualitatively different.

---

> > > ### Author Response · Authors · 2024-11-19
> > > **Formulas used in experiments**
> > >
> > > In the context of our experiments, all prior knowledge can be easily expressed using a CNF of reasonable size (ie. polynomial in the number of variables):
> > > - Categorical constraints can be expressed as a CNF as shown on Example 3 of the paper.
> > > - Hierarchical formulas can be expressed as CNF as shown on Example 4 of the paper.
> > > - Simple path constraints on a directed graph G:=(V, E) can be expressed using clauses that ensures that: an outgoing edge of the source node s is true, if an incoming edge of a node is true then an outgoing edge of this node is also true, two outoing edges of the same node cannot be true simultaneously:
> > > $$ \kappa_G := (\bigvee_{e \in E_{out}(s) Y_e}) \land  \bigwedge_{v \in V, e \in E_{in}(v)} (\neg Y_e \lor_{w \in E_{out}(v)} Y_w ) \land \bigwedge_{v \in V, u, w \in E_{out}(v), u \neq w} (\neg Y_u \lor \neg Y_w) $$
> > >
> > > Thus, conditioned distributions for all these types of constraints can be efficiently captured by a PGM.
> > >
> > > Besides on hierarchical tasks tackled in the paper, the tree-width of the particular formulas is rather low, hence we can even use PGM message passing algorithms to perform MPE and PQE. Although custom algorithms might be more efficient.
> > >
> > > However, there is a problem when dealing with simple path constraints, since these constraints can easily have unbounded tree-width. For instance in the case of kxk grids, as used in our experiments, the treewidth of the CNF formula that captures the simple path semantics is in O(k). Therefore, standarg PGM message passing algorithms are not efficient for performing MPE and PQE in such cases. This means that the PGM representation, although possible, becomes less useful.

---

### Official Review · Reviewer_Ydn5 · 2024-11-04

**Soundness:** 4
**Presentation:** 4
**Contribution:** 2
**Rating:** 5
**Confidence:** 4

**Summary:**

The paper presents a unified formalism that encapsulates learning and inference in the presence of prior knowledge specified in propositional logic, also known as neurosymbolic AI. Using the presented formalism, they are able to delineate the approaches present
in the literature, and define a new technique which they call *semantic conditioning at inference*. The proposed method applies to the
neural network only during inference, and does not impose any changes during training.

**Strengths:**

- The paper is pretty well written, and provides a comprehensive overview of the different neurosymbolic approaches in the literature

- The proposed approach, *semantic conditioning at inference*, greatly boosts the accuracy of the models on the tasks considered
in the experimental section, while at the same time avoiding the computational costs of performing MAP inference on the full distribution.

**Weaknesses:**

- While the paper provides a nice, unifying overview of the field and the key techniques developed, I am not quite sure that such a view is especially novel

- Furthermore, and perhaps more crucial, is that other work such as that by Niepert et al and Pogancic et al, as well as other works using combinatorial solvers do exactly what the authors set out to do: compute the MPE state at inference time without maintaining the distribution over the entire space of outputs, thereby also avoiding the often prohibitive task of counting, and which the authors do not compare to. This is my biggest concern, as it puts into question the novelty of the authors' technical contributions.

- In the experimental section, I initially found it hard to parse the Figure 2 due to the absence of an explicit mention of what the acronyms used stand for.

**Questions:**

Please see the weaknesses section.

---

> ### Author Response · Authors · 2024-11-14
> **Comparison to Niepert et al. and Pogancic et al.**
>
> The comparison to Nierpert et al. and Pogancic et al. can be delicate since their work is not mainly centered on informed multi-label classification, as in our paper, but tries to integrate combinatorial solvers inside a differentiable pipeline of neural networks. However, their work can be adapted into a neurosymbolic technique for informed classification, as illustrated by their experiments on simple path tasks, and it is of this adaption that we talk below.
>
> The originality of Nipert's work is that they do not rely on exact counting but on approximate counting using the Gumbel trick with perturbed calls to exact MPE solvers. In other terms, their technique can be seed as an approximate method for semantic conditioning when PQE cannot be solved exactly but MPE can. Therefore, if it is similar to our contribution (semantic conditioning at inference) in terms of computational complexity in the sense that they do not rely on solving exact PQE, even though several calls to MPE solvers are necessary in their case while only one is used in semantic conditioning at inference. However, it also impacts training and is closer to semantic conditioning than semantic conditioning at inference in this sense (which is not surprising since it is an approximation of semantic conditioning). This implies that it cannot be applied to usecases based on foundation or off-the-shelves models like semantic conditioning at inference can.
>
> Moreover, some of their usecases rely on a restriction on the space of logits (for instance only allowing positive logits) to ensure the tractability of MPE, which is not the case in our work.
>
> Since exact semantic conditioning could be implemented tractably for all the tasks considered in our experiments, we decided not to include the approximate version allowed by the work of Niepert and Pogancic. We agree that a comparison with their work on tasks where semantic conditioning at inference was prefered to semantic conditioning for computational complexity purposes would be very useful.
>
> NB: I am not sure to understand what you mean by "maintaining the distribution over the entire space of outputs". If you mean that the value of the distribution is not computed for all possible states of the output space, then it is the case of all probabilistic neurosymbolic techniques, including those that perform weighted counting (PQE), as such an implementation would very quickly become intractable.

---

> > ### Comment · Reviewer_Ydn5 · 2024-11-26
> >
> > Thank you for your response and apologies for the late reply.
> >
> > I agree that a large contribution of the work by Niepert et al. and Pogancic et al. is how to differentiate through a combinatorial solver. Essentially, they proposed a method to further improve upon semantic conditioning at inference: they not only return the MPE, which is what semantic conditioned at inference does, but they also adapt the weights of the neural network to be able to better solve the task. But ultimately, given a test example, they do perform semantic conditioning at inference. Therefore, I believe at least of discussion of how one can expect the contributions of the current paper to differ from that line work is in order.
> >
> > RE: "maintaining the distribution over the entire space of outputs", I just meant computing Probabilistic Query Estimation, as done by all the exact probabilistic neuro-symbolic techniques.

---

### Meta-Review · Area_Chair_ET2U · 2024-12-20

**Metareview:**

The paper proposes a framework for encoding prior knowledge within probabilistic neurosymbolic approaches for multi-label classification.
While in existing work, semantic conditioning is applied both at training time and at inference time, the authors propose to do this only at inference time.

The reviewers collectively found the paper well-written and appreciated the attempt at unification, but remained concerned about the incremental nature of the contribution, the limited set of baselines, and the lack of thorough comparative analysis. The authors failed to provide a detailed rebuttal and offer clarifications. Given these persistent concerns, we recommend rejecting this paper. We believe that this paper would be a strong one after addressing these concerns.

**Additional Comments On Reviewer Discussion:**

During the rebuttal phase, the reviewers raised several main concerns, and the authors failed to provide sufficient responses. Specifically:

1. **Incremental technical Contribution (Raised by Reviewer Ydn5, Reviewer yzJ2, Reviewer 4WvM):**
   Several reviewers questioned the novelty of the proposed “semantic conditioning at inference” (SCI). They noted that SCI might be seen as an ablation of existing methods (full semantic conditioning) rather than a fundamentally new technique claimed by authors.


2. **Comparison to Related Work (Raised by Reviewer Ydn5, Reviewer  mFsY):**
   Some reviewers felt the paper lacked sufficient comparison with closely related methods in the neurosymbolic domain, particularly those that also leverage maximum probable explanation or approximate methods at inference time. Others suggested stronger connections to probabilistic graphical models and related literature in multi-label classification, where constraints can be encoded similarly.


3. **Baselines and Evaluation (Raised by Reviewer 4WvM, Reviewer mFsY):**
   Reviewers requested more robust baselines, including comparisons against fuzzy regularization or other advanced multi-label classification techniques, rather than just “uninformed” models. They also suggested conducting ablation studies and analyzing the distribution of constraint violations or error cases. More thorough quantitative and qualitative evaluations (e.g., what kind of mistakes SCI makes vs. full semantic conditioning) were desired.

4. **Clarity, Background, and Practicality (Raised by Reviewer yzJ2, Reviewer mFsY):**
   Some reviewers felt the paper focused too heavily on background and formal definitions at the expense of practical insights. They wanted clearer examples, a better explanation of constraints used in experiments, and a more direct demonstration of why a practitioner should choose SCI over existing methods.

---

### Decision · Program_Chairs · 2025-01-22

Reject